# Can morphological features of coccolithophores serve as a reliable proxy to reconstruct environmental conditions of the past?

Giulia Faucher[1], Ulf Riebesell[2], Lennart Thomas Bach[3]

[1]Dipartimento di Scienze della Terra "Ardito Desio", Università degli Studi di Milano, Milano, 20133 Italia
[2]Biological Oceanography, GEOMAR Helmholtz Centre for Ocean Research Kiel, Kiel 24105, Germany
[3]Institute for Marine and Antarctic Studies, University of Tasmania, Hobart, Tasmania, Australia

*Correspondence to*: Giulia Faucher (giulia.faucher@unimi.it)

**Abstract.** Morphological changes in coccoliths, tiny calcite platelets covering the outer surface of coccolithophores, can be induced by physiological responses to environmental changes. Coccoliths recovered from sedimentary successions may therefore provide information on paleo-environmental conditions prevailing at the time when the coccolithophores were alive. To calibrate the biomineralization responses of ancient coccolithophore to environmental changes, studies often compared the biological responses of living coccolithophore species with paleo-data from calcareous nannofossils. However, there is uncertainty whether the morphological responses of living coccolithophores are representative for those of the fossilized ancestors. To investigate this, we exposed four living coccolithophore species (*Emiliania huxleyi, Gephyrocapsa oceanica, Coccolithus pelagicus* subsp. *braarudii,* and *Pleurochrysis carterae)* that have been evolutionarily distinct for hundred thousand to millions of years, to a range of environmental conditions (i.e. changing light intensity, Mg/Ca ratio, nutrient availability, temperature and carbonate chemistry) and evaluated their responses in coccolith morphology (i.e. size, length, width, malformation). The motivation for this study was to test if there is a consistent morphological response of the four species to changes in any of the tested abiotic environmental factors. If this was the case, then this could suggest that coccolith morphology can serve as a paleo-proxy for that specific factor because this response is conserved across species that have been evolutionary distinct over geological timescales. However, we found that the four species responded differently to changing light intensity, Mg/Ca ratio, nutrient availability and temperature in terms of coccolith morphology. The lack of a common response reveals the difficulties in using coccolith morphology as a paleo-proxy for these environmental drivers. However, a common response was observed under changing seawater carbonate chemistry (i.e. rising $CO_2$) which consistently induced malformations. This commonality provides some confidence that malformations found in the sedimentary record could be indicative for adverse carbonate chemistry conditions.

# 1 Introduction

Coccolithophores are calcifying marine phytoplankton and are among the most important calcifiers in the ocean (Tyrell and
Young, 2010). They produce single calcitic platelets named coccoliths and nannoliths. Due to their ability to calcify,
coccolithophores played an important role in rock-formation during the Jurassic and Cretaceous as well as through the
Cenozoic (e.g. Erba, 2006). They are directly affected by environmental drivers such as temperature, salinity, nutrient
concentration, light, and carbonate chemistry that can modify physiological rates and morphology of certain taxa (e.g. Paasche,
1998; Riebesell et al., 2000; Langer et al., 2006; Trimborn et al., 2007; Zondervan et al., 2007; Rosas-Navarro et al., 2016).
Due to their sensitivities to environmental changes fossil remains of coccolithophores (coccoliths and nannoliths), have often
been used as paleo-proxies to reconstruct past physical and chemical conditions in the surface ocean of local or global
significance (e.g. Erba, 1994; Lees et al., 2005; Tiraboschi et al., 2009; Erba et al., 2010; Lübke and Mutterlose, 2016; Faucher
et al., 2017a; Erba et al., 2019; supplementary S1). To calibrate the biomineralization responses of ancient coccolithophorid
algae to climatic changes, studies often compared the biological responses of living coccolithophore species with paleo-data
from calcareous nannofossils (Table 1).

The primary goal of our study was to understand if physiological experiments with contemporary species are a valid tool to
predict responses of ancient coccolithophores to environmental change in the geological record. The assumption that modern
species respond identically to environmental change than ancient species, is implicit in many studies (e.g. Giraud et al., 2006;
Erba et al., 2010; Faucher et al., 2017a, Table 1) but, to the best of our knowledge, not been explored in depth. To test this
assumption, we did a series of identical stress test experiments with four selected modern species that have been evolutionarily
distinct since hundred thousand to millions of years (Fig. 1). Our hypothesis was: in case that coccolith morphology responses
to a changing environmental driver are similar in the four species, this could be indicative of a response pattern that was
physiologically conserved over geological timescales because the species were evolutionarily separated for so long. In other
words, if species conserve a similar response to certain types of environmental change for geological timescales, despite very
different evolutionary trajectories, then this would strengthen our confidence that responses recorded for modern species also
apply for the geological past.

Indeed, there is considerable uncertainty when trying to reconstruct paleo-environmental conditions based on coccolith
morphology. This in itself is not surprising considering that there are millions of years of evolution between the time when the
fossil coccolithophores lived and when the physiological experiments were done (Bown, 2005; De Vargas et al., 2007).
Moreover, for the fossil record, it is extremely difficult to disentangle the individual factor(s) that drove changes in coccolith
morphology. Therefore, it is unsurprising that studies occasionally come to different conclusions about what environmental
factor drove a morphological change in the paleo-record. For example, Erba et al. (2010), detected the reduction in size and
variation in shape of some nannofossil species during a time of excess volcanogenic $CO_2$ emissions. They explained their trend
with detrimental carbonate chemistry conditions based on physiological incubation studies by Riebesell et al. (2000) who
found decreasing calcification rates under increasing $CO_2$. Conversely, Bornemann and Mutterlose (2006) explained

decreasing coccolith size with decreasing sea surface temperature, a conclusion that was also based on incubation experiments with living coccolithophore species (Renaud and Klaas, 2001; Renaud et al., 2002).

In order to investigate our hypothesis outlined above, we selected four different coccolithophore species: *Emiliania huxleyi* (morphotype R)*, Gephyrocapsa oceanica*, *Coccolithus pelagicus* subsp. *braarudii,* and *Pleurochrysis carterae*. According to "molecular-clock-data", they are evolutionarily distinct since the Triassic or the Jurassic (with the exception of *G. oceanica* and *E. huxleyi* that diverged ~ 290 *Kya;* Liu et al., 2010; Bendif et al., 2014). We present data on how coccolith size and morphology change in response to a suite of different environmental drivers and explore whether there is a common response to any of these drivers among the different species. Afterwards, we discuss if morphological features of coccoliths have the potential to serve as paleo-proxies.

## 2 Material and Methods

### 2.1 Experimental setup

Five experiments are presented in this study with a similar design. Every experiment tested the influence of one abiotic parameter on four different coccolithophore species which were cultured individually (i.e. in separate bottles). The tested abiotic factors were: light intensity, nutrient limitations (N or P limitations), Mg/Ca ratio, temperature, carbonate chemistry. Monospecific cultures of the coccolithophores *Emiliania huxleyi* (strain RCC 1216, from the Tasmanian sea), *Gephyrocapsa oceanica* (strain RCC 1303, from the France coast of the Atlantic Ocean), *Coccolithus pelagicus* subsp. *braarudii* (strain PLY182G, it will be called hereafter *C. braarudii*, from the English Channel, Atlantic Ocean), and *Pleurochrysis carterae* (unknown strain number, coastal species) were grown in artificial seawater (Kester et al., 1967) under dilute batch culture conditions (LaRoche et al., 2010). The artificial seawater medium was enriched with 64 µmol kg$^{-1}$ nitrate, 4 µmol kg$^{-1}$ phosphate to avoid nutrient limitations with the exception of the nutrient limitation experiment (see section 2.1.3). In all experiments we added f/8 concentrations of vitamins and trace metals (Guillard and Ryther, 1962), 10 nmol kg$^{-1}$ of SeO$_2$ (Danbara and Shiraiwa, 1999) and 2 ml kg$^{-1}$ of natural North Sea water to provide potential nutrients which were not added with the nutrient cocktail (Bach et al., 2011). The medium was sterile-filtered (0.2 µm). The carbonate chemistry was adjusted with aeration for 24 h using a controlled CO$_2$ gas mixing system reaching the treatment levels of 400 (total alkalinity, TA, 2302 µmol kg$^{-1}$) with the exception of the carbonate chemistry experiment (see section 2.1.5).

The medium was then transferred into 0.5 L NalgeneTM bottles. Cultures were incubated in a thermo constant climate chamber (Rubarth Apparate GmbH) at a constant temperature of 15 °C, (with the exception of the temperature experiment; see section 2.1.4), with a 16:8 [hour:hour] light/dark cycle, at a photon flux density of 150 µmol photons m$^{-2}$s$^{-1}$ (with the exception of the light experiment; see section 2.1.1). Before the beginning of the experiments, coccolithophore cultures were acclimated for about 7-10 generations to each of the experimental conditions. Cultures were in the exponential growth phase at the initiation of the experiments (also in the nutrient limitation experiment; see section 2.1.3). All culture bottles were manually and carefully rotated three times a day, each time with 20 rotations in order to reduce sedimentation bias. Final samples were taken when

cells were exponentially growing (except for nutrient limitation experiments; see section 2.1.3) but cell numbers were still low enough to limit their influence on the chemical conditions of the growth medium. Sampling was conducted at the same time for every experiment to avoid changes in cell diameter/volume which develop in light-dark cycles due to the synchrony of the cell cycle (Müller et al., 2012; Sheward et al., 2017).

### 2.1.1 Specifics in the light experiment

The light setup was adjusted to test the response of the four species to a gradient of photon flux densities (PFD). Because light intensities are difficult to replicate we chose a gradient design in this experiment at the expense of replication (Cottingham et al., 2005). Therefore, the light was set to the highest possible intensity in the light chamber and the bottles were placed at different positions so that 12 different PFDs were established (50, 100, 150, 200, 250, 300, 350, 400, 450, 500, 550, 600 µmol photons $m^{-2}s^{-1}$). Light intensities were measured at every treatment position in the light chamber, using a Li-250A light meter (Li-Cor, Heinz Walz GmbH, Effeltrich).

### 2.1.2 Specifics in the Mg/Ca experiment

This experiment was designed to test the physiological response of coccolithophore algae to changing $[Ca^{2+}]$ while keeping $[Mg^{2+}]$ constant at the modern seawater value. In the control, the Mg/Ca ratio was set to simulate the modern ocean values (Mg/Ca = 5.2) with $[Ca^{2+}] = 9.8$ mmol $L^{-1}$ and $[Mg^{2+}] = 50$ mmol $L^{-1}$. The low Mg/Ca treatments were set by increasing $[Ca^{2+}]$ to 25 and 50 mmol $L^{-1}$, respectively. The control and both treatments were replicated three times.

### 2.1.3 Specifics in the nutrient experiment

Batch cultures were grown under N or P limitations. For N-limitation, all cultures were run into N-limitation during the acclimation phase but care was taken that this occurred at low cell densities so that the chemical conditions in the seawater (apart from nutrients) remained largely unaffected. During the main experiment, cell concentrations were counted every other day and 0.14 pmol N $cell^{-1}$ (as $NaNO_3$) was added to the medium when cultures reached the stationary phase (i.e. they stopped dividing). The same was done in P-limitation experiments except that 0.01 pmol P $cell^{-1}$ (as $NaH_2PO_4$) was added when reaching the stationary phase. As control, we used exponentially growing cells which were replete in both N and P. Nutrient concentrations were not measured but limitations were assured by measuring and comparing growth rates which were much lower than in the nutrient replete controls. Controls and both treatments were replicated three times.

### 2.1.4 Specifics in the temperature experiment

The experiments were carried out in two temperature-controlled light chambers in order to test the response of the coccolithophores to increased temperature. Batch cultures were grown at 15 °C and 22.5 °C. Both temperature treatments were replicated three times.

### 2.1.5 Specifics in the carbonate chemistry experiment

In the ocean acidification (OA) treatment, TA was kept constant (2348 µmol kg$^{-1}$) whereas fCO$_2$ was increased to 1020.5 µatm. In the Cretaceous scenario1 (CS1) treatment, fCO$_2$ was kept constant at 1020.5 µatm, while TA was increased to 3729 µmol kg$^{-1}$. In the Cretaceous scenario2 (CS2) treatment, fCO$_2$ was increased up to 3061 µatm and TA up to 4978 µmol kg$^{-1}$. Carbonate chemistry parameters (pHf (free scale), HCO$_3^-$, CO$_3^{2-}$, CO$_2$) were calculated using the program CO2SYS (Pierrot et al., 2006) from measured TA, and calculated estimated DIC, temperature, salinity and [PO$_4$], and the dissociation constants determined by Roy et al. (1993). In the OA, CS1 and CS2 treatments, DIC and TA levels were adjusted by adding calculated amounts of Na$_2$CO$_3$ (Merck, Suprapur quality and dried for 12 hours at 500°C) and hydrochloric acid (3.571 mol L$^{-1}$, certified by Merck) following Gattuso et al. (2010).

Samples for pH and TA analyses were taken at the beginning and at the end of the experiments. Samples were filtered (0.7 µm) and stored at 4 °C until measurements that were performed within 2 days for pH measurements and 14 days for TA. pH was measured spectrophotometrically with Varian Cary 100 in 10 cm cuvette at 25 °C as described in Dickson et al. (2007) and then recalculated to in-situ temperature (15 °C) using CO2SYS as is described by Schulz et al. (2017). Every sample was measured 3 times. Samples for TA were measured in duplicate with Metrohm 862 Compact Tritino Sampler device following Dickson (2003). TA data were accuracy controlled with certified reference material (A. Dickson, La Jolla, CA).

### 2.2 Cell abundance, coccosphere and cell size

Samples for cell abundance were taken at the end of the experiment with the exception of the nutrient experiments where samples were taken every second day. Incubation bottles were turned to resuspend all cells and to obtain a homogenous suspension of the cells before sampling. Cell numbers were immediately measured three times without addition of preservatives using a Beckman coulter Multisizer. After the abundance measurements, samples were acidified with 0.1 mmol L$^{-1}$ HCl to dissolve all free and attached coccoliths and subsequently measured another 3 times each in order to obtain cell diameters and volumes (Müller et al., 2012).

### 2.3 Scanning Electron Microscopy (SEM)

Samples for SEM analysis were filtered by gravity onto polycarbonate filters (0.2 µm pore size). For every sample, 5-10 ml of water was used. Filters were subsequently dried at 60°C for two days. Samples were sputtered with gold-palladium. SEM analysis was performed at the Earth Sciences department of the University of Milan with SEM Cambridge Stereoscope 360. All pictures were taken with the same magnification (5000x) and the scale bar given on SEM pictures was used for calibration. For every experiment, in all treatments and replicates, 50 specimens for each species were analyzed. For every coccolith, the length (DSL) and the width (DSW) of the coccolith distal shield were manually measured using the public domain program Fiji distributed by ImageJ software (Schindelin et al., 2012). For *E. huxleyi*, the inner tube thickness, the number distal shield elements, and the distal shield elements thickness were also measured. For *G. oceanica* the tube thickness and the bridge

orientations were measured. Moreover, the presence of malformations was quantified by visual inspection (Fig. 2): morphologies were grouped following Langer et al. (2006) and Langer et al. (2010) categories.

**2.4 Statistics**

Data were tested for normality and homogeneity of variances (Bartlett and Fligner-Killeen tests). To test the null hypothesis that differences in growth rates and sizes among treatments are the same, the average values of parameters from triplicate cultures were compared between treatments. A one-way analysis of variance was used to determine the statistical significance of the main effect of the different parameters tested on the variables. A Tukey post-hoc test was used to assess whether

differences between treatments or control were statistically significant. Statistical treatments of data were performed using R software. Statistical significance was accepted for $p < 0.05$. For the light experiment, a non-linear regression was used to explore the relationship between light and coccolithophore parameters (growth and sizes).

**3 Results**

**3.1 Light**

In the four species selected, coccolithophore, cell, and coccolith sizes didn't show any distinct trend with variable light intensity. Data are reported in Table 2. *Emiliania huxleyi* coccoliths were less elliptical with light intensities above 400 μmol photons $m^{-2}$ $s^{-1}$ and characterized by a higher number of distal shield elements with light intensities above 400 μmol photons $m^{-2}$ $s^{-1}$. *Gephyrocapsa oceanica* and *C. braarudii* coccolith size and shape did not change with light intensity. Finally, *P. carterae* coccoliths were less elliptical only at irradiances of 350 μmol photons $m^{-2}$ $s^{-1}$. (Fig. 3; supplementary, plate 1).

Malformed coccoliths increased in percentage only in *E. huxleyi* at 500 μmol photons $m^{-2}$ $s^{-1}$ and in *G. oceanica* at 200 μmol photons $m^{-2}$ $s^{-1}$ (Fig. 4).

**3.2 Mg/Ca**

*Emiliania huxleyi,* coccosphere, and cell sizes were influenced by changes in seawater $[Ca^{2+}]$. Elevating seawater $[Ca^{2+}]$ to $\approx$ 25 mmol $L^{-1}$ and 50 mmol $L^{-1}$ resulted in a significant increase in the coccosphere and cell diameters ($p < 0.05$). Increased

$[Ca^{2+}]$ concentrations impacted *G. oceanica, C. braarudii* and *P. carterae* cell sizes with a reduction in size in comparison to $[Ca^{2+}]$ of 9.8 mmol $L^{-1}$, when seawater $[Ca^{2+}]$ was elevated to $\approx$ 25 mmol $L^{-1}$ and 50 mmol $L^{-1}$. *Gephyrocapsa oceanica* and *P. carterae* coccosphere diameters were unaffected while the *C. braarudii* coccosphere was smaller when grown under $[Ca^{2+}]$ of 50 mmol $L^{-1}$ (Table 3). *Emiliania huxleyi, G. oceanica,* and *C. braarudii* coccolith sizes were not affected by changing $[Ca^{2+}]$. *Pleurochrysis carterae* coccoliths were smaller at the highest $[Ca^{2+}]$ concentrations than in the control (Fig. 3; Table 3).

*Emiliania huxleyi* produced a higher percentage of malformed and/or incomplete coccoliths with increasing calcium concentrations (Fig. 4; supplementary, plate 2) while no increased malformation was observed in the other species.

### 3.3 Nutrient limitation

*Emiliania huxleyi* and *C. braarudii* coccospheres were larger under P-limitation than under N-limitation and the control. *Gephyrocapsa oceanica* coccospheres were larger under N-limitation than under P-limitation and the control. *Pleurochrysis carterae* coccospheres were larger under N-limitation compared to the control. Cell size remained unaffected in *E. huxleyi* by nutrient limitation. *Gephyrocapsa oceanica* cell and *C. braarudii* cells were larger under P-limitation compared to the control and N limitation. *Pleurochrysis carterae* cells were larger under N limitation compared to the control (Table 4).

*Emiliania huxleyi* and *G. oceanica* coccoliths were larger under P limitation, while there was no significant difference between N limitation and the control. *Emiliania huxleyi* coccoliths had a higher number of distal shield elements under P limitations while the inner tube was thinner in N and P limited treatments compared to the control. *Gephyrocapsa oceanica* produced thicker inner tubes under N and P limitation. *Coccolithus braarudii* was less elliptical under P limitation and *P. carterae* was less elliptical under N limitation. Furthermore, *E. huxleyi* and *G. oceanica* produced relatively more malformed coccoliths under P limitation (Fig. 4). *Coccolithus braarudii* and *P. carterae* coccolith sizes remained unaffected with no sign of malformations by nutrient limitation (Fig. 3; supplementary, plate 3).

### 3.4 Temperature

*Emiliania huxleyi* and *G. oceanica* coccospheres and cell sizes were smaller at 22.5 °C. *Pleurochrysis carterae* coccosphere and cell sizes remained unaffected (Table 5). *Emiliania huxleyi* coccoliths were smaller at high temperatures. Furthermore, *E. huxleyi* had less distal shield elements and a thinner inner tube when grown at 22.5°C. *Gephyrocapsa oceanica* and *P. carterae* coccolith size remained largely unaffected by changing temperature but *G. oceanica* produced thicker inner tubes under high temperature. *Pleurochrysis carterae* coccoliths were less elliptical when grown at 22.5°C (Fig. 3; Table 5; supplementary, plate 3). *Coccolithus braarudii* did not survive at 22.5°C condition.

### 3.5 Carbonate chemistry parameters

*Emiliania huxleyi* coccospheres and cells were the largest in the OA treatment and smallest in the CS2 treatment. *Gephyrocapsa oceanica* and *C. braarudii* coccospheres were the largest in the control and smallest in CS2 treatment. *Gephyrocapsa oceanica* cell size was lower in the CS2 treatment than in the control, as well as the OA and CS1 treatments. The cell size of *C. braarudii* was smaller in the OA, CS1 and CS2 treatments compared to the control. *Pleurochrysis carterae* coccosphere and cell size were unaffected by changing carbonate chemistry (Table 6).

*Emiliania huxleyi* formed significantly bigger coccoliths in the OA treatment compared to the control and the CS2 treatment (Fig. 3; Table 6). Furthermore, the inner tubes were thicker in the OA and CS1 treatments compared to the control and the CS2 treatments. Malformations were 20% more frequent in the OA, CS1 and CS2 treatments than in the control (Fig. 4; supplementary, plate 4). *Gephyrocapsa oceanica* generated a high number of malformed coccoliths in the OA and CS2 treatments. For *G. oceanica*, under OA and CS2 conditions, morphometric analyses were not performed because a large

majority of the coccoliths were extremely malformed and it wasn't possible to measure the shape of the specimens (Fig. 2). In the CS1 treatment, coccoliths were slightly smaller compared to the control with a thinner inner tube. *Coccolithus braarudii*

coccoliths were smaller in the OA and CS2 treatments compared to the control and the CS1. In the OA and CS1 treatments, 40% of the *C. braarudii* coccoliths were malformed and ~ 10 %  were incomplete. In the CS2 treatment, 97% of coccoliths were malformed or incomplete. *Pleurochrysis carterae* coccolith size remained unaffected by carbonate chemistry variations but coccoliths are less elliptical under OA, CS1, and CS2 compared to the control.

## 4 Discussion

Coccolithophores started to calcify in the late Triassic and this biological innovation appeared in a period of strong climatic and biotic pressure (De Vargas et al., 2007). The earliest coccoliths had very simple morphologies and small sizes (2-3 µm; Bown et al., 2004). Calcareous nannoplankton underwent a major diversification in the Mesozoic and Paleocene where many new morphologies occurred. The appearance of new coccolith shapes followed the main geological events, at the K/Pg boundary, and the P/E boundary, and these big reorganizations suggest that certain kinds of morphologies might have been no

longer advantageous for coccolithophore algae under the new ecological circumstances. The evolution of calcareous nannoplankton through ~220 Ma documents a remarkable morphological diversity within the group and in the last 30 Ma there has been a loss of species that produced large and heavily calcified coccoliths but an increase in the modern community of coccolith architectures (Bown et al., 2004). The cause of this impressive number of structures is unknown but there might be a reason connected to the function of coccoliths for the different species to produce such different shapes ranging from

protection against excess sun light and/or against grazing (Monteiro et al., 2016). Accordingly, coccolith morphologies are likely only indirectly linked to physical or chemical conditions such as temperature or $CO_2$ but may rather reflect their adaptation to a specific, yet unknown ecological function (Aloisi, 2015). If morphological changes in coccoliths are the result of a physiological response to environmental variations (e.g. $CO_2$, nutrient, temperature), coccoliths recovered from marine sediments could potentially conserve paleo-environmental information prevailing when the coccolithophore was alive (Aloisi,

2015). Indeed, many studies on geological records calibrated biomineralization responses of ancient species to environmental drivers with experiments with modern species (e.g. Bornemann et al., 2006; Erba et al., 2010; Suchéras-Marx et al., 2010; Linnert and Mutterlose, 2012; O'Dea et al., 2014; Lübke et al., 2015; Gibbs et al., 2016; Faucher et al., 2017a; Faucher et al., 2017b; Table 1). Calibrating paleo-responses with observations from living species depends on the assumption that coccolithophores conserved a certain response to certain environmental parameters over geological timescales. However,

fossils and living coccolithophores diverged a long time ago, have a different genetic background and therefore, calcareous nannoplankton in the past and nowadays did and do not necessarily act in the same way to external stress. Furthermore, morphology may not only depend on abiotic environmental conditions but could perhaps also be the result of evolutionary development induced through ecological interactions. For example, if a coccolithophore genotype which forms larger coccoliths is better suited to protect a cell against prevalent grazers then these genotypes will likely proliferate whereas related

genotypes forming smaller variants could eventually go extinct. The geological record would not easily allow us to distinguish if morphological changes are caused by physiological or ecological drivers as it is difficult enough to reconstruct abiotic paleo-environmental conditions but almost impossible to unravel relevant processes in the food web of the geologic past. Therefore, the fundamental question we asked ourselves was whether morphological features observed in living coccolithophores under specific environmental parameters could help to build reliable proxies for abiotic paleo-environmental conditions.

Overall, none of the five tested variable*s* induced a consistent response of coccolith size and shape across all four species. For example, under high $CO_2$, *E. huxleyi* formed larger coccoliths while *C. braarudii* formed smaller coccoliths. Interestingly, our observation of inconsistent responses among species to various environmental drivers is in line with observations from the fossil record. There are several observations where just some of the prevailing species showed changes in morphology during intervals characterized by extreme climatic conditions even though all species were exposed to environmental stress. For

example, dwarf specimens were recorded for *Biscutum constans* in all Mesozoic episodes characterized by abnormal conditions, during intervals of extreme volcanic activity (e.g. during Oceanic Anoxic Event (OAE) 1a, OAE 1b, OAE 2; Bornemann et al., 2006; Erba et al., 2010; Lübcke et al., 2015; Faucher et al., 2017a; Erba et al., 2019). In conclusion, the inconsistency of morphological responses to changing environmental drivers observed in both our experiments and the geological record suggests that morphological responses of living species cannot be used as analogues for morphological

changes of extinct species.

The exception in our dataset is the observed responses in malformation to changes in carbonate chemistry where some consistency was noted among the four tested species. Malformations are generally considered as an evidence of errors during intracellular coccolith formation so that a disturbance of coccolithogenisis conserved in a malformation could be the consequence of a direct (i.e. physiological) impact. Indeed, malformations are unlikely to be the consequence of an

evolutionary (i.e. ecological) adaptation to environmental stress because there seems to be no obvious ecological advantage of producing malformed coccoliths. The high degree of malformation when coccolithophores were grown under high $CO_2$ concentrations provides some evidence that at least this response variable could be used as paleo-proxy for episodes of acute carbonate chemistry perturbations.

In the fossil record, there are several examples of intervals characterized by high abundances of malformed specimens, linked

to the low calcite saturation state of the ocean (Jiang and Wise, 2006; Raffi and De Bernardi, 2008; Agnini et al., 2007; Erba et al., 2010; Bralower and Self Trail, 2016). Different authors argued for high $CO_2$ influence on causing these malformations during the Mesozoic OAEs, Paleocene-Eocene Thermal Maximum (PETM) and Eocene Thermal Maximum 2. All these intervals were characterized by excess $CO_2$ concentrations and/or slightly reduced pH. Malformations were expressed in different ways: they were represented by variation in ellipticity of coccoliths (Erba et al., 2010), asymmetry (Agnini et al,

2007), irregular arrangement and length of their rays, and diminished calcification in some nannoliths (Jiang and Wise, 2006; Mutterlose et al., 2007; Raffi and De Bernardi, 2008; Bralower and Self Trail, 2016). The short stratigraphic ranges where these malformations occurred, during the core of major ocean perturbations, indicated that pH played a role in inducing the production of these aberrant specimens (Mutterlose et al., 2007; Erba et al., 2010). There is still not a clear explanation of why

only some species of calcareous nannoplankton were producing aberrant specimens, and there is not a general consensus on the role of carbonate chemistry on coccolithophore biomineralization (Gibbs et al., 2010; Gibbs et al., 2016). However, a recent work provides a plausible explanation of what might have happened during the PETM. Here, only some species moved and inhabited the deep part of the photic zone, to possibly refuge from stressful warm and eutrophic conditions of the surface water, but had to deal with lower saturation conditions that induced malformations in these taxa (Bralower and Self Trail, 2016). The increase in the percentage of malformed coccoliths observed in our experiments could suggest a more global occurrence of malformation in modern coccolithophore species under low pH. However, it is important to bear in mind that in the geological record critical intervals characterized by excess $CO_2$ concentrations lasted for some tens or hundred thousand years, whereas our experiments lasted a few generations (days). Thus, environmental stress on geological timescales may still be long enough for coccolithophores to adapt which can occur within months to years (Lohbeck et al., 2012; Bach et al., 2018). It also needs to be kept in mind that even if the four coccolithophore species tested here showed similar morphological responses to changing environmental drivers, it cannot be excluded that this resulted from convergent evolution. Indeed, restriction on biological conditions and adaptation to particular habitats can produce widespread convergence as convergent evolution is often a consequence of adaptation to a similar niche (Arbuckle et al., 2014). Therefore, we want to point out that convergent morphological developments could represent similar adaptations of different species to abiotic parameters that occurred multiple times separate from each other.

**4 Conclusions**

According to the data provided in this study we report the following results: 1) sizes and morphologies of the four tested species change differently in response to temperature, light, nutrient, and Mg/Ca variations. In some cases, there were opposing reactions among species under the same abiotic stress; 2) A high number of malformations were detected when coccolithophores were grown under excess $CO_2$ and this response occurred in all species tested here.

Overall, there is no support for the suitability of coccolith morphometry to serve as proxy for temperature, light, nutrient, and Mg/Ca conditions of the past. However, coccolith malformations could perhaps be useful indicators for carbonate chemistry stress. Indeed, it will be crucial to evaluate whether malformations remain over long time period or if coccolithophores have and had an adaptive potential towards extreme carbonate chemistry conditions that might rapidly eliminate malformation in some generations.

The authors declare that they have no conflict of interest

**Acknowledgments**

We are indebted to the Editor Erin McClymont, to Mariem Saavedra-Pellitero and an anonymous reviewer for their fruitful comments that greatly improved the quality of this manuscript. A special note of thank goes to Elisabetta Erba for sharing ideas and huge support. We acknowledge Agostino Rizzi for assistance during the never-ending SEM analyses. This research
was funded through MIUR-PRIN 2011 (Ministero dell'Istruzione, dell'Università e della Ricerca–Progetti di Ricerca di Interesse Nazionale) to Elisabetta Erba and through SIR-2014 (Ministero dell'Istruzione, dell'Università e della Ricerca– Scientific Independence of young researchers) to Cinzia Bottini.

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

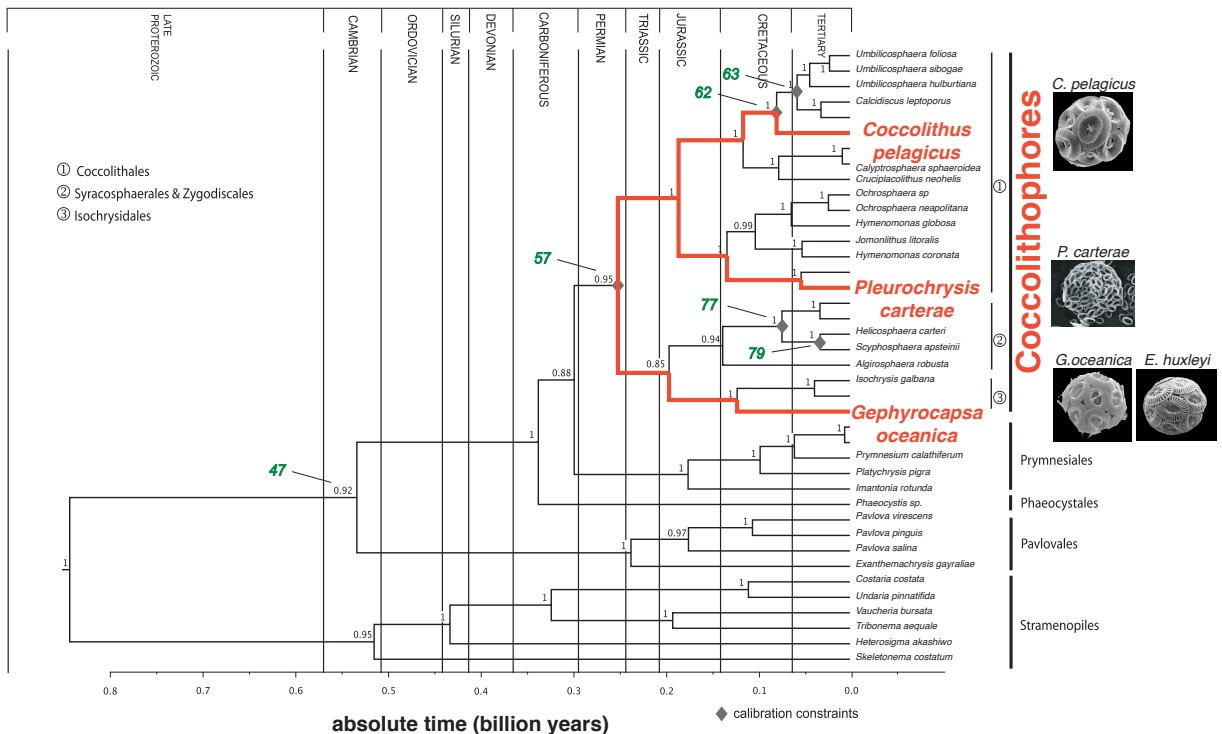

**Figure 1: Phylogeny and divergence times of the Haptophytes, modified from Liu et al. (2010). Time is indicated in billion years. The species selected for this study are shown in red. The nodes represent following divergence episodes. The number in green represents specific nodes: node 47:** *Exanthemachrysis gayraliae* **and** *Helicosphaera carteri*; **node 57:** *Coccolithus pelagicus* **and** *H. carteri*; **node 62:** *C. pelagicus* **and** *Umbilicosphaera hulburtiana*; **node 63:** *Calcidiscus leptoporus* **and** *Umbilicosphaera foliosa*; **node 77:** *Coronosphaera mediterranea* **and** *Scyphosphaera apsteinii*; **node 79:** *H. carteri* **and** *S. apsteinii* **(node 79). Numbers are related to calculated divergence times. For further information see Liu et al. 2010.**



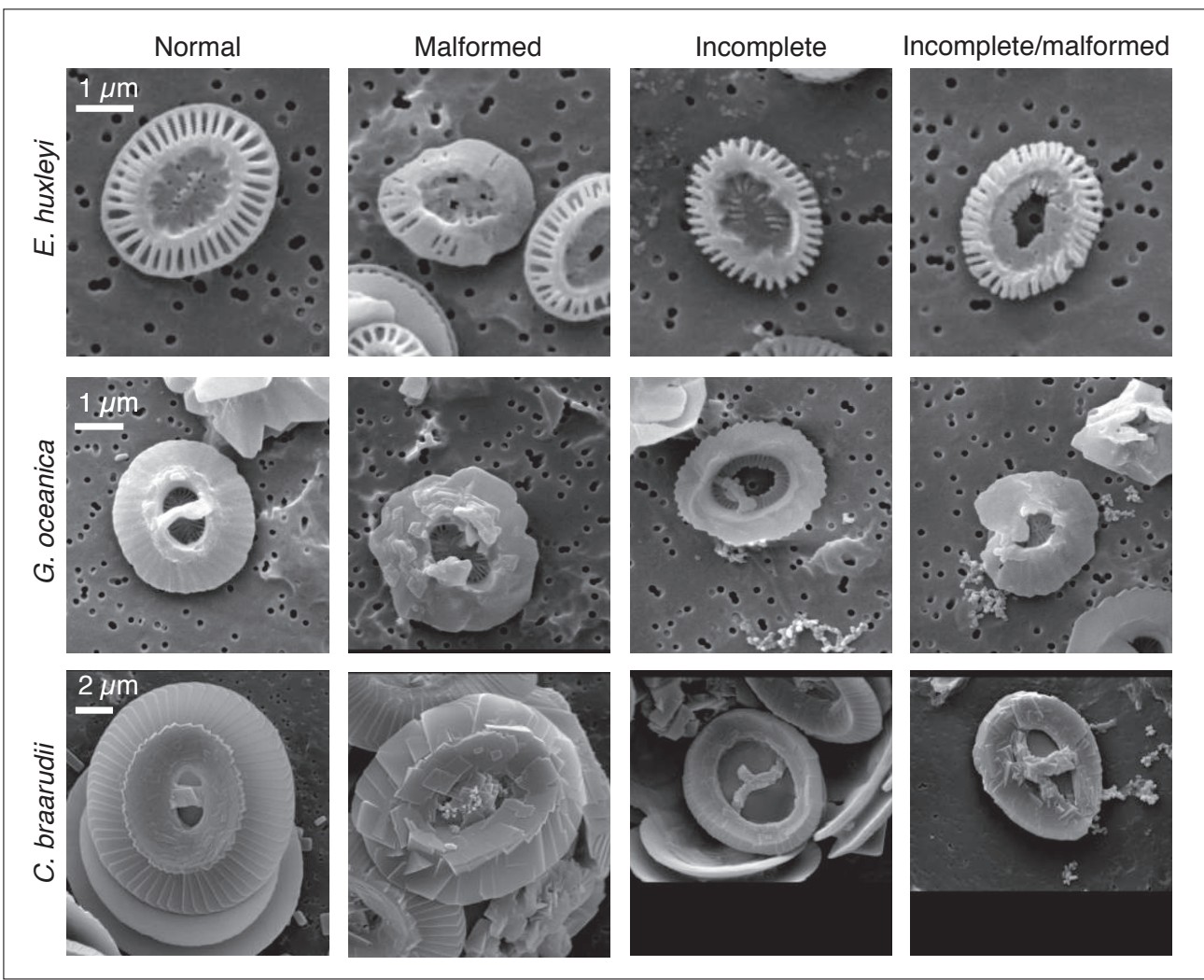

**Figure 2: Examples of different morphological categories: normal, malformed, incomplete and incomplete/malformed for *E. huxleyi, G. oceanica* and *C. braarudii*.**



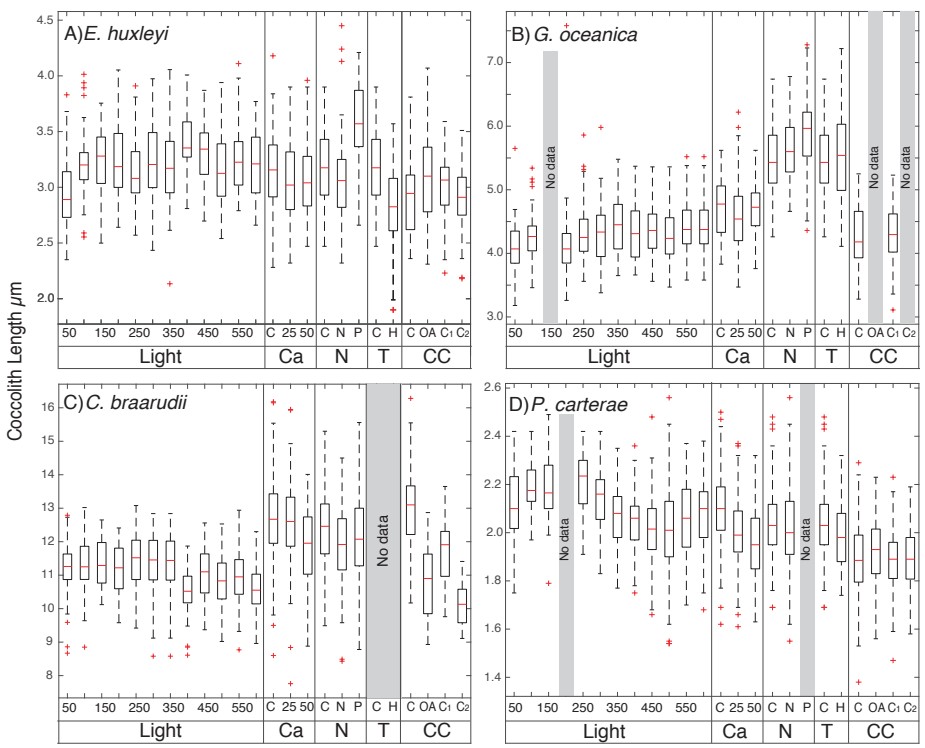

**Fig. 3 Box plots of coccolith length from the different experiments. In A)** *E. huxleyi*; **B)** *G. oceanica*; **C)** *C. braarudii*; **D)** *P. carterae*. **C= control treatment for every experiment. Light: experiment with 12 different light intensities from 50 to 600 µm photons m$^{-2}$ s$^{-1}$; Ca: calcium manipulation experiment, 25, $[Ca^{2+}]$ =25 mmol L$^{-1}$; 50, $[Ca^{2+}]$= 50 mmol L$^{-1}$. N: nutrient limitation experiment, N= nitrogen limited condition; P= phosphate limited condition. T: temperature experiment, H = 22.5 °C; CC: carbonate chemistry experiment; theoretical CO$_2$ values: C= 400 ppm, OA, ocean acidification=1000 ppm, C1, cretaceous scenario1= 1000 ppm, C2 cretaceous scenario2= 3000 ppm (for further information see paragraphs 2.1.5 and 3.5). The tops and bottoms of each "box" are the 25th and 75th percentiles of the samples respectively. The red line in the middle of each box is the median. The whiskers, extending above and below each box, represent the furthest observations. Observations beyond the whisker length are marked as outliers (red cross). For the light experiment, 50 specimens were considered for every treatment. For Mg/Ca experiment (Ca), nutrient experiment (N), temperature (T) and carbonate chemistry manipulations (CC) experiments, every box plot represents 150 measurements in total, (50 measurements for each replicate). The Light experiment was performed in December 2013; the Ca experiment was performed in June 2014; the N experiment was performed in December 2017; the T experiment was performed in October 2017; the CC experiment was performed in August 2014.**



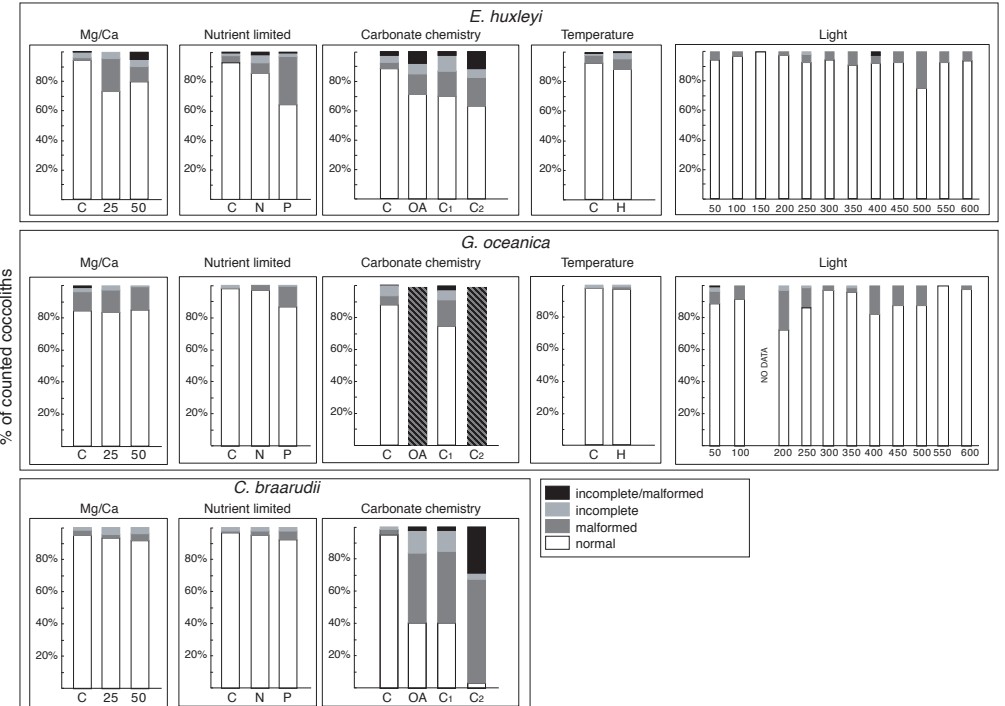

**Figure 4: Percentage of normal, malformed, incomplete and incomplete/malformed coccoliths versus experiments. The experiments displayed represent Mg/Ca, nutrient limitation, carbonate chemistry, temperature, and light intensity manipulations.** *C. braarudii* **didn't survive at high temperature (22.5 °C) and no malformations were observed under the different light intensities tested; therefore, percentage of malformations are not represented for these experiments for this species. Furthermore, no malformation was observed for** *P. carterae* **and percentage are not shown. C= control treatment. Mg/Ca: calcium manipulation experiment, 25, $[Ca^{2+}]$ =25 mmol $L^{-1}$; 50, $[Ca^{2+}]$= 50 mmol $L^{-1}$; Nutrient limited: nutrient limitation experiment, N= nitrogen limited condition; P= phosphate limited condition. Temperature experiment, H = 22.5 °C; Carbonate chemistry experiment; theoretical $CO_2$ values: C= 400 ppm, OA, ocean acidification=1000 ppm, C1, cretaceous scenario1= 1000 ppm, C2 cretaceous scenario2= 3000 ppm. Light: 12 different light intensities from 50 to 600 μm photons $m^{-2}$ $s^{-1}$. For every treatment and for every replicate 100 specimens were considered.**


| | Paper | Analyzed genera or species | morphological variations | implicated environmental parameter | Age | Investigated sections or cores |
|---|---|---|---|---|---|---|
| **Malformations** | Agnini et al. 2006 | calcareous nannoplankton assemblage referred to as the Calcareous Nannoplankton Excursion Taxa (CNET) | malformations of CNET | high $CO_2$ and transient chemical modifications of the world's oceans | Paleocene–Eocene Thermal Maximum (PETM) | Possagno section, Italy |
| | Agnini et al. 2007 | calcareous nannoplankton assemblage referred to as the Calcareous Nannoplankton Excursion Taxa (CNET) | high asymmetry of CNET | high $CO_2$, low pH, change in temperature structure | PETM | Forada section, Italy |
| | Bralower and Self Trail 2016 | *Discoaster* | irregularity of individual rays or whole *Discoaster* | high $CO_2$ | PETM | Bass River, Wilson Lake, South Dover Bridge cores, USA |
| | Erba et al. 2010 | *Biscutum constans, Discorhabdus rotatorius, Zeugrabdotus erectus, Watznaueria barnesiae* | malformations of *Watznaueria barnesiae* | high $CO_2$, and low pH | Aptian Oceanic Anoxic Event (OAE) 1a | Cismon core, Italy; DSDP site 463 |
| | Jiang and Wise 2006 | *Discoaster* | malformed *Discoaster araneus* and *Discoaster anartios;* | low pH | PETM | ODP Site 1259 |
| | Mutterlose et al. 2007 | *Discoaster araneus* | asymmetrical, aberrant *Discoaster araneus* | high $CO_2$, and/or low pH, and/or warming, increased nutrients | PETM | ODP Site 1260B |
| | Raffi and De Bernardi 2008 | *Fasciculithus, Discoaster nobilis* group, *Discoaster mediosus*, and *Discoaster multiradiatus* | malformed *Discoaster nobilis, Discoaster falcatus* and *Discoaster mediosus* and weakly calcified fasciculith specimens (*F. thomasii* morphotype) | high $CO_2$ | PETM | ODP Site 1263 |
| **Size variations** | Bornemann et al. 2003 | *Watznaueria* spp., *Conusphaera mexicana, Polycostella beckmannii* and *Nannoconus* spp. | small *Watznaueria* | changes of climate and the circulation pattern | latest Tithonian and earliest Berriasian | DSDP Sites 105, 534A, 367 |
| | Bornemann and Mutterlose 2006 | *Biscutum constans, Watznaueria barnesiae* | reduced *Biscutum constans* size | cooler water conditions | Late Albian | Col de Palluel section, France |
| | Erba et al. 2010 | *Biscutum constans, Discorhabdus rotatorius, Zeugrabdotus erectus, Watznaueria barnesiae* | dwarf *Biscutum constans, Discorhabdus rotatorius* and *Zeugrabdotus erectus* | high $CO_2$, and low pH | OAE 1a | Cismon core, Italy; DSDP site 463 |
| | Ferreira et al. 2017 | seven morphospecies of *Lotharingius* | smaller morphotypes | unstable environmental conditions: seawater temperature fluctuations, water trophic variations and expansion/contraction of the photic zone and nutricline | Toarcian - early Aalenian | Rabaçal, Brenha, Cabo Mondego sections, Portugal; Truc-de-Balduc section, France |
| | Faucher et al. 2017 | *Biscutum constans, Discorhabdus rotatorius, Zeugrabdotus erectus, Watznaueria barnesiae* | size reduction of *Biscutum constans* | high $CO_2$ and high trace metal concentrations | Late Cenomanian and latest Cenomanian OAE 2 | Clot Chevalier section, France; Eastbourne section, United Kingdom; Novara di Sicilia section, Italy; Cuba and Pueblo sections, USA |
| | Fraguas and Young 2011 | genus *Lotharingius* | "dwarfing'' of *Lotharingius hauffii, Lotharingius sigillatus* and *Lotharingius cruicentrali* | increased temperature and perturbation of the carbon cycle | Early Toarcian | West Rodiles section, Spain |
| | Giraud et al. 2006 | *Watznaueria britannica* | bigger *Watznaueria britannica* | oligotrophic conditions and warmer climate | Late Oxfordian- early Kimmeridgian | Balingen–Tieringen section, Germany |
| | Linnert and Mutterlose 2012 | genera *Biscutum, Broinsonia, Prediscosphaera, Retecapsa* and *Watznaueria* | reduction in size of *Biscutum* and *Broinsonia* spp. | reduced nutrient availability (*Biscutum*), increase in sea-surface temperature (*Broinsonia* spp.) | Cenomanian–Turonian | Goban Spur cores Site 549, Site 551, Germany |
| | Lübke et al. 2015 | *Biscutum constans, Zeugrhabdotus erectus* and *Watznaueria barnesiae* | small *Biscutum constans* | low seawater temperatures, low light availability, high nutrient level | OAE 1a | North Jens-1, Adda-2, Alstätte1 cores, Germany; Cismon core, Italy; DSDP Leg 62; |
| | Mattioli et al. 2004a | genera *Biscutum* and *Similiscutum* | bigger *Biscutum* and *Similiscutum* | high seawater temperature and/or to high nutrient concentration | Pliensbachian-Toarcian | DSDP 547B; Monte Genuardo, Somma section, Italy; Dotternhausen, section Germany |
| | Mattioli et al. 2004b | *Schizosphaerella* | small *Schizosphaerella* and undercalcified coccoliths | increased $CO_2$ | early Toarcian OAE | Pozzale and Somma sections, Italy; Dotternhausen section, Germany; Brown Moor Borehole, United Kingdom |
| | Mattioli et al. 2009 | *Schizosphaerella* | small *Schizosphaerella* | increased $CO_2$ or temperature rise or less saline marine surface waters | Early Toarcian OAE | Dotternhausen section, Germany; Somma section, Italy; Peniche section Portugal; HTM-102 borehole, Saint Paul de Fonts section, France |
| | O'Dea et al. 2014 | *Coccolithus pelagicus* | thinning of *Coccolithus pelagicus* coccoliths | ocean acidification | PETM | Bass River, Lodo Gulch, USA; Tanzania Drilling Project Site 14, Tanzania |
| | Suan et al. 2008 | *Schizosphaerella* | small *Schizosphaerella* | increased $CO_2$ and high seawater temperature | Early Toarcian OAE | Peniche section, Portugal |
| | Suan et al. 2010 | *Schizosphaerella* | small *Schizosphaerella* | high nutrient concentrations (less oligotrophic conditions) and/or increased temperature | Early Toarcian | Peniche section, Portugal |
| | Suchéras-Marx et al. 2010 | *Crepidolithus crassus* | variation in pseudo-cryptic species dominance, alternation of "small" and " big" *Crepidolithus crassus* | fluctuation of the nutricline and photic zone depth under the control of the orbital cycles of eccentricity and precession | Early Pliensbachian | Peniche section, Portugal |
| | Tremolada et al. 2008 | *Discoaster multiradiatus* | migration of allochtonus specimens of larger *Discoaster multiradiatus* | increased seawater temperature, stratification of water masses and establishment of a well-defined thermocline | PETM | ODP Sites 690 and 1209 |
| | Wulff et al. 2020 | *Biscutum constans, Rhagodiscus asper* and *Watznaueria barnesiae* | small *Biscutum constans* | oligotrophic surface water conditions. | Barremian | "Frielingen" 9 core, road cut "A39" motorway, Braunschweig, Germany |

**Table 1 Compilation of papers documenting morphological (malformation) or morphometrical variations of calcareous nannofossils. We considered papers that compared shape and size variations of calcareous nannoplankton in the fossil record with biological responses of living coccolithophores from culture experiments. Papers are organized in alphabetic order, the following information is given: authors, analyzed species, morphological and morphometrical variations, environmental parameter/s considered for the detected alteration, age, and investigated cores/sections.**








| Light | | | | | | | | | | | tube thick. | |
|---|---|---|---|---|---|---|---|---|---|---|---|---|
| *E. huxleyi* | µ | coccosphere | cell | DSL | d.StdL | DSW | d.StdW | Ellipticity | SE | SEW | tube thick. | |
| 50 | 0.41 | 4.57 | 3.94 | 2.95 | 0.33 | 2.43 | 0.33 | 1.22 | 29.3 | 0.11 | 0.37 | |
| 100 | 0.43 | 4.57 | 4.02 | 3.20 | 0.28 | 2.59 | 0.23 | 1.23 | 30.5 | 0.12 | 0.32 | |
| 150 | 0.74 | 4.82 | 4.52 | 3.23 | 0.30 | 2.64 | 0.27 | 1.23 | 31.3 | 0.10 | 0.29 | |
| 200 | 0.59 | 5.05 | 4.46 | 3.24 | 0.33 | 2.66 | 0.29 | 1.22 | 30.5 | 0.11 | 0.31 | |
| 250 | 1.01 | 5.01 | 4.41 | 3.19 | 0.30 | 2.63 | 0.26 | 1.22 | 32.2 | 0.11 | 0.32 | |
| 300 | 1.03 | 4.87 | 4.50 | 3.25 | 0.34 | 2.68 | 0.32 | 1.22 | 33.0 | 0.11 | 0.32 | |
| 350 | 1.06 | 5.05 | 4.39 | 3.21 | 0.35 | 2.64 | 0.31 | 1.22 | 32.7 | 0.11 | 0.36 | |
| 400 | 1.18 | 5.02 | 4.43 | 3.39 | 0.33 | 2.81 | 0.30 | 1.21 | 33.1 | 0.10 | 0.33 | |
| 450 | 1.20 | 5.03 | 4.42 | 3.30 | 0.28 | 2.74 | 0.26 | 1.21 | 33.5 | 0.11 | 0.38 | |
| 500 | 1.10 | 5.01 | 4.38 | 3.20 | 0.34 | 2.64 | 0.31 | 1.21 | 32.7 | 0.10 | 0.35 | |
| 550 | 0.97 | 4.85 | 4.35 | 3.27 | 0.30 | 2.72 | 0.27 | 1.21 | 33.2 | 0.11 | 0.33 | |
| 600 | 0.87 | 4.92 | 4.34 | 3.20 | 0.30 | 2.65 | 0.27 | 1.21 | 33.2 | 0.11 | 0.33 | |
| *G. oceanica* | µ | coccosphere | cell | DSL | d.StdL | DSW | d.StdW | Ellipticity | | | tube thick. | angle ° |
| 50 | 0.52 | 6.78 | 5.17 | 4.10 | 0.39 | 3.55 | 0.36 | 1.16 | | | 1.34 | 57.41 |
| 100 | 0.66 | 7.75 | 5.55 | 4.27 | 0.39 | 3.69 | 0.38 | 1.16 | | | 1.32 | 61.68 |
| 200 | 0.63 | 7.22 | 5.23 | 4.17 | 0.65 | 3.61 | 0.53 | 1.16 | | | 1.34 | 61.69 |
| 250 | 0.67 | 6.83 | 5.47 | 4.32 | 0.48 | 3.76 | 0.46 | 1.15 | | | 1.35 | 61.88 |
| 300 | 0.67 | 6.74 | 5.27 | 4.34 | 0.53 | 3.71 | 0.51 | 1.17 | | | 1.40 | 65.91 |
| 350 | 0.71 | 6.74 | 5.16 | 4.46 | 0.46 | 3.90 | 0.41 | 1.14 | | | 1.32 | 58.12 |
| 400 | 0.66 | 6.88 | 5.51 | 4.33 | 0.45 | 3.84 | 0.40 | 1.13 | | | 1.23 | 66.67 |
| 450 | 0.74 | 6.92 | 5.60 | 4.36 | 0.43 | 3.76 | 0.43 | 1.16 | | | 1.29 | 65.11 |
| 500 | 0.63 | 6.57 | 4.88 | 4.25 | 0.40 | 3.71 | 0.34 | 1.15 | | | 1.26 | 62.89 |
| 550 | 0.71 | 6.62 | 4.97 | 4.38 | 0.46 | 3.82 | 0.43 | 1.15 | | | 1.27 | 61.29 |
| 600 | 0.50 | 6.41 | 4.92 | 4.29 | 0.45 | 3.72 | 0.42 | 1.16 | | | 1.25 | 68.27 |
| *C. braarudii* | µ | coccosphere | cell | DSL | d.StdL | DSW | d.StdW | Ellipticity | | | | |
| 50 | 0.40 | 18.74 | 13.35 | 11.24 | 0.87 | 9.34 | 0.83 | 1.21 | | | | |
| 100 | 0.35 | 17.58 | 10.82 | 11.31 | 0.82 | 9.47 | 0.71 | 1.20 | | | | |
| 150 | 0.42 | 17.81 | 10.72 | 11.35 | 0.72 | 9.46 | 0.73 | 1.20 | | | | |
| 200 | 0.44 | 19.37 | 11.26 | 11.19 | 0.75 | 9.38 | 0.72 | 1.19 | | | | |
| 250 | 0.44 | 17.80 | 11.33 | 11.36 | 0.81 | 9.63 | 0.80 | 1.18 | | | | |
| 300 | 0.49 | 17.55 | 10.94 | 11.41 | 0.88 | 9.54 | 0.75 | 1.20 | | | | |
| 350 | 0.52 | 17.54 | 10.94 | 11.35 | 0.80 | 9.58 | 0.78 | 1.19 | | | | |
| 400 | 0.52 | 17.35 | 10.20 | 10.49 | 0.71 | 8.88 | 0.66 | 1.18 | | | | |
| 450 | 0.49 | 18.60 | 12.33 | 11.00 | 0.74 | 9.42 | 0.67 | 1.17 | | | | |
| 500 | 0.49 | 17.69 | 10.52 | 10.81 | 0.82 | 9.13 | 0.72 | 1.19 | | | | |
| 550 | 0.50 | 17.59 | 10.62 | 10.94 | 0.75 | 9.20 | 0.70 | 1.19 | | | | |
| 600 | 0.63 | 17.19 | 10.95 | 10.55 | 0.73 | 8.91 | 0.80 | 1.19 | | | | |
| *P. carterae* | µ | coccosphere | cell | DSL | d.StdL | W | d.StdW | Ellipticity | | | | |
| 50 | 0.18 | 11.02 | 7.93 | 2.12 | 0.13 | 1.32 | 0.08 | 1.61 | | | | |
| 100 | 0.19 | 11.60 | 8.91 | 2.19 | 0.10 | 1.36 | 0.06 | 1.62 | | | | |
| 150 | 0.17 | 11.41 | 8.69 | 2.18 | 0.12 | 1.35 | 0.09 | 1.62 | | | | |
| 250 | 0.19 | 12.16 | 10.52 | 2.22 | 0.18 | 1.40 | 0.14 | 1.61 | | | | |
| 300 | 0.29 | 13.09 | 10.28 | 2.14 | 0.13 | 1.34 | 0.09 | 1.60 | | | | |
| 350 | 0.28 | 12.73 | 10.32 | 2.08 | 0.12 | 1.33 | 0.11 | 1.57 | | | | |
| 400 | 0.28 | 11.84 | 10.48 | 2.05 | 0.13 | 1.35 | 0.10 | 1.58 | | | | |
| 450 | 0.28 | 11.35 | 10.73 | 2.02 | 0.15 | 1.31 | 0.10 | 1.56 | | | | |
| 500 | 0.28 | 11.73 | 10.15 | 1.97 | 0.23 | 1.26 | 0.17 | 1.58 | | | | |
| 550 | 0.26 | 12.71 | 9.97 | 2.04 | 0.15 | 1.30 | 0.10 | 1.58 | | | | |
| 600 | 0.32 | 12.06 | 9.82 | 2.07 | 0.13 | 1.32 | 0.12 | 1.58 | | | | |

**Table 2 Light experiments data. Growth rate (µ, cell d$^{-1}$); coccosphere, cell diameters (µm); coccolith morphometric analysis were performed on 50 specimens for every treatment: average of coccolith distal shield length (DSL; µm) and coccolith distal shield width (DSW; µm); ellipticity (L/W); average *E. huxleyi* distal shield elements number (SE) and average distal shield elements width (µm, SEW); average *E. huxleyi* inner tube thickness (µm, tube thick); *G. oceanica* tube thickness (µm, tube thick.); *G. oceanica* bridge angle (angle °). For *G. oceanica* and *P. carterae*, data from 150 and 200 µmol photons m$^{-2}$s$^{-1}$ are missing due to errors on light intensity inside the light cabinet; dStd= standard deviation.**



| E. huxleyi | μ | d.Std | coccosphere | d.Std | cell | d.Std | DSL | d.StdL | DSW | d.StdW | Ellipticity | SE | SEW | tube thick. | |
|---|---|---|---|---|---|---|---|---|---|---|---|---|---|---|---|
| Control | 0.97 | 0.01 | 4.69 | 0.01 | 4.08 | 0.00 | 3.14 | 0.32 | 2.57 | 0.29 | 1.22 | 30 | 0.13 | 0.37 | |
| [Ca$^{2+}$] = 25 mmol L$^{-1}$ | 0.92 | 0.01 | 4.84 | 0.06 | 4.22 | 0.03 | 2.99 | 0.33 | 2.44 | 0.28 | 1.20 | 30 | 0.11 | 0.35 | |
| [Ca$^{2+}$] = 50 mmol L$^{-1}$ | 0.85 | 0.00 | 4.92 | 0.02 | 4.33 | 0.02 | 3.07 | 0.34 | 2.50 | 0.30 | 1.23 | 31 | 0.11 | 0.35 | |
| G. oceanica | μ | d.Std | coccosphere | d.Std | cell | d.Std | DSL | d.StdL | DSW | d.StdW | Ellipticity | | | tube thick. | angle ° |
| Control | 0.65 | 0.00 | 6.86 | 0.09 | 5.29 | 0.03 | 4.72 | 0.48 | 4.14 | 0.50 | 1.15 | | | 1.43 | 66.49 |
| [Ca$^{2+}$] = 25 mmol L$^{-1}$ | 0.60 | 0.01 | 6.82 | 0.04 | 5.16 | 0.01 | 4.61 | 0.53 | 4.01 | 0.46 | 1.15 | | | 1.40 | 68.27 |
| [Ca$^{2+}$] = 50 mmol L$^{-1}$ | 0.55 | 0.00 | 6.83 | 0.04 | 5.15 | 0.03 | 4.73 | 0.38 | 4.18 | 0.40 | 1.14 | | | 1.46 | 71.75 |
| C. braarudii | μ | d.Std | coccosphere | d.Std | cell | d.Std | DSL | d.StdL | DSW | d.StdW | Ellipticity | | | | |
| Control | 0.56 | 0.01 | 19.82 | 0.11 | 15.65 | 1.39 | 12.66 | 1.29 | 11.00 | 1.21 | 1.16 | | | | |
| [Ca$^{2+}$] = 25 mmol L$^{-1}$ | 0.39 | 0.01 | 19.44 | 0.19 | 12.08 | 0.39 | 12.57 | 1.27 | 10.79 | 1.19 | 1.17 | | | | |
| [Ca$^{2+}$] = 50 mmol L$^{-1}$ | 0.48 | 0.01 | 18.74 | 0.09 | 11.59 | 0.40 | 11.88 | 1.16 | 10.63 | 0.72 | 1.16 | | | | |
| P. carterae | μ | d.Std | coccosphere | d.Std | cell | d.Std | DSL | d.StdL | DSW | d.StdW | Ellipticity | | | | |
| Control | 0.35 | 0.04 | 11.63 | 0.22 | 9.56 | 0.21 | 2.10 | 0.14 | 1.34 | 0.10 | 1.57 | | | | |
| [Ca$^{2+}$] = 25 mmol L$^{-1}$ | 0.39 | 0.04 | 11.09 | 0.15 | 8.73 | 0.03 | 1.93 | 0.14 | 1.30 | 0.09 | 1.54 | | | | |
| [Ca$^{2+}$] = 50 mmol L$^{-1}$ | 0.41 | 0.01 | 11.39 | 0.34 | 9.00 | 0.16 | 2.00 | 0.18 | 1.24 | 0.12 | 1.56 | | | | |

**Table 3 Mg/Ca experiment data. Data presented are the average of three replicates. Growth rate (μ, cell d$^{-1}$); coccosphere, cell diameters (μm); coccolith morphometric analysis were performed on 50 specimens for every treatment and for every replicate. Data represent the average of three replicates: average of coccolith distal shield length (DSL; μm) and coccolith distal shield width (DSW; μm); ellipticity (L/W) diameter; average *E. huxleyi* distal shield elements number (SE) and average distal shield elements width (μm, SEW); average *E. huxleyi* inner tube thickness (μm, tube thick.), *G. oceanica* bridge angle (angle °); dStd= standard deviation.**




| E. huxleyi | μ | d.Std | coccosphere | d.Std | cell | d.Std | DSL | d.StdL | DSW | d.StdW | Ellipticity | SE | SEW | tube thick. | |
|---|---|---|---|---|---|---|---|---|---|---|---|---|---|---|---|
| Control | 0.94 | 0.01 | 4.94 | 0.07 | 4.28 | 0.03 | 3.18 | 0.31 | 2.61 | 0.28 | 1.22 | 32 | 0.12 | 0.48 | |
| N limited | 0.40 | 0.02 | 5.08 | 0.00 | 4.17 | 0.23 | 3.05 | 0.33 | 2.49 | 0.28 | 1.23 | 31 | 0.12 | 0.37 | |
| P limited | 0.31 | 0.06 | 7.28 | 0.00 | 3.97 | 0.18 | 3.57 | 0.35 | 2.96 | 0.32 | 1.21 | 35 | 0.12 | 0.41 | |
| G. oceanica | μ | d.Std | coccosphere | d.Std | cell | d.Std | DSL | d.StdL | DSW | d.StdW | Ellipticity | | | tube thick. | angle ° |
| Control | 0.63 | 0.07 | 8.01 | 0.14 | 6.29 | 0.09 | 5.47 | 0.54 | 4.66 | 0.48 | 1.18 | | | 1.48 | 64.58 |
| N limited | 0.15 | 0.09 | 10.27 | 0.40 | 8.21 | 0.70 | 5.63 | 0.45 | 4.75 | 0.41 | 1.19 | | | 1.54 | 63.80 |
| P limited | 0.36 | 0.02 | 9.20 | 0.75 | 7.51 | 0.53 | 5.91 | 0.60 | 5.03 | 0.53 | 1.18 | | | 1.60 | 65.06 |
| C. braarudii | μ | d.Std | coccosphere | d.Std | cell | d.Std | DSL | d.StdL | DSW | d.StdW | Ellipticity | | | | |
| Control | 0.54 | 0.01 | 18.86 | 0.03 | 11.48 | 0.14 | 12.37 | 1.11 | 10.60 | 0.97 | 1.17 | | | | |
| N limited | 0.10 | 0.00 | 19.83 | 0.10 | 10.69 | 0.09 | 11.92 | 1.16 | 10.11 | 1.05 | 1.18 | | | | |
| P limited | 0.32 | 0.02 | 28.89 | 2.02 | 13.53 | 0.94 | 12.10 | 1.14 | 10.58 | 1.11 | 1.15 | | | | |
| P. carterae | μ | d.Std | coccosphere | d.Std | cell | d.Std | DSL | d.StdL | DSW | d.StdW | Ellipticity | | | | |
| Control | 0.44 | 0.04 | 11.38 | 0.25 | 8.94 | 0.64 | 2.04 | 0.14 | 1.27 | 0.10 | 1.60 | | | | |
| N limited | 0.20 | 0.04 | 13.15 | 0.54 | 11.01 | 0.73 | 2.01 | 0.18 | 1.29 | 0.13 | 1.57 | | | | |

**Table 4 Nutrient limited condition experiment data. Data presented are the average of three replicates. Growth rate (μ, cell d⁻¹);** — **coccosphere, cell diameters (μm); coccolith morphometric analysis were performed on 50 specimens for every treatment and for every replicate. Data represent the average of three replicates: average of coccolith distal shield length (DSL; μm) and coccolith distal shield width (DSW; μm); ellipticity (L/W) diameter; average *E. huxleyi* distal shield elements number (SE), and average distal shield elements width (μm, SEW); average *E. huxleyi* inner tube thickness (μm, tube thick.), *G. oceanica* bridge angle (angle°);**

**dStd= standard deviation.**

| E. huxleyi | μ | d.Std | coccosphere | d.Std | cell | d.Std | DSL | d.StdL | DSW | d.StdW | Ellipticity | SE | SEW | tube thick. | |
|---|---|---|---|---|---|---|---|---|---|---|---|---|---|---|---|
| Control | 0.94 | 0.01 | 4.94 | 0.07 | 4.28 | 0.03 | 3.18 | 0.31 | 2.61 | 0.28 | 1.22 | 32 | 0.12 | 0.48 | |
| High | 1.47 | 0.01 | 3.70 | 0.10 | 3.18 | 0.03 | 2.82 | 0.34 | 2.33 | 0.29 | 1.21 | 29 | 0.11 | 0.27 | |
| G. oceanica | μ | d.Std | coccosphere | d.Std | cell | d.Std | DSL | d.StdL | DSW | d.StdW | Ellipticity | | | tube thick. | angle ° |
| Control | 0.63 | 0.07 | 8.01 | 0.14 | 6.29 | 0.09 | 5.47 | 0.54 | 4.66 | 0.48 | 1.18 | | | 1.48 | 64.58 |
| High | 1.11 | 0.08 | 7.21 | 0.06 | 5.51 | 0.02 | 5.55 | 0.66 | 4.74 | 0.62 | 1.17 | | | 1.59 | 67.65 |
| P. carterae | μ | d.Std | coccosphere | d.Std | cell | d.Std | DSL | d.StdL | DSW | d.StdW | Ellipticity | | | | |
| Control | 0.44 | 0.04 | 11.38 | 0.25 | 8.94 | 0.64 | 2.04 | 0.14 | 1.27 | 0.10 | 1.60 | | | | |
| High | 0.31 | 0.04 | 11.82 | 0.23 | 9.05 | 0.23 | 1.99 | 0.13 | 1.37 | 0.10 | 1.45 | | | | |

**Table 5 Temperature experiment data. Data presented are the average of three replicates. Growth rate (μ, cell d$^{-1}$) coccosphere, cell diameters (μm); coccolith morphometric analysis were performed on 50 specimens for every treatment and for every replicate. Data represent the average of three replicates: average of coccolith distal shield length (DSL; μm) and coccolith distal shield width (DSW; μm); ellipticity (L/W) diameter; average *E. huxleyi* distal shield elements number (SE), and average distal shield elements width (μm, SEW); average *E. huxleyi* inner tube thickness (μm, tube thick.), *G. oceanica* bridge angle (angle°). *C. braarudii* didn't grow at 22.5°C and therefore, any data is presented, dStd= standard deviation.**

**E. huxleyi**

|  | Control | OA | CS1 | CS2 |
|---|---|---|---|---|
| pH | 7.99 | 7.65 | 7.81 | 7.53 |
| TA | 2302 | 2302 | 3611 | 4931 |
| fCO2 | 460 | 1068 | 1178 | 3142 |
| HCO3- | 1958.58 | 2114 | 3289 | 4714 |
| Ca out | 3.50 | 1.74 | 3.88 | 2.93 |
| μ | 1.05 | 0.76 | 1.12 | 0.52 |
| dev. Std | 0.03 | 0.08 | 0.11 | 0.00 |
| coccosphere | 4.88 | 5.09 | 4.94 | 4.7 |
| dev. Std | 0.11 | 0.11 | 0.08 | 0.01 |
| cell | 4.23 | 4.59 | 4.35 | 4.44 |
| dev. Std | 0.24 | 0.01 | 0.03 | 0.00 |
| DSL | 2.93 | 3.09 | 3.01 | 2.89 |
| d.StdL | 0.33 | 0.37 | 0.3 | 0.29 |
| DSW | 2.39 | 2.53 | 2.44 | 2.38 |
| d.StdW | 0.28 | 0.33 | 0.27 | 0.24 |
| ellipticity | 1.23 | 1.23 | 1.24 | 1.21 |
| SE | 28 | 33 | 30 | 30 |
| SEW | 0.11 | 0.11 | 0.12 | 0.1 |
| tube thick. | 0.25 | 0.29 | 0.31 | 0.23 |

**G. oceanica**

|  | Control | OA | CS1 | CS2 |
|---|---|---|---|---|
| pH | 7.91 | 7.79 | 7.79 | 7.53 |
| TA | 2172 | 2303 | 3611 | 4933 |
| fCO2 | 570 | 1366 | 1256 | 3142 |
| HCO3- | 1945 | 2164 | 3303 | 4714 |
| Ca out | 3.01 | 1.52 | 3.74 | 2.93 |
| μ | 0.66 | 0.27 | 0.57 | 0.15 |
| dev. Std | 0.00 | 0.01 | 0.04 | 0.00 |
| coccosphere | 7.25 | 6.24 | 6.51 | 5.44 |
| dev. Std | 0.01 | 0.12 | 0.25 | 0.01 |
| cell | 5.45 | 5.40 | 5.31 | 4.83 |
| dev. Std | 0.06 | 0.13 | 0.03 | 0.01 |
| DSL | 4.28 | No data | 4.32 | No data |
| d.StdL | 0.46 |  | 0.47 |  |
| DSW | 3.73 |  | 3.71 |  |
| dev. Std | 0.42 |  | 0.40 |  |
| ellipticity | 1.15 |  | 1.17 |  |
|  |  |  |  |  |
| tube thick. | 1.32 |  | 1.23 |  |
| angle | 63.81 |  | 70.29 |  |

**C. braarudii**

|  | Control | OA | CS1 | CS2 |
|---|---|---|---|---|
| pH | 7.86 | 7.60 | 7.78 | 7.51 |
| TA | 2170 | 1994 | 3234 | 4895 |
| fCO2 | 591 | 1142 | 1190 | 3356 |
| HCO3- | 1800 | 2009 | 3123 | 4711 |
| Ca out | 2.68 | 1.52 | 3.54 | 2.77 |
| μ | 0.56 | 0.43 | 0.42 | 0.23 |
| dev. Std | 0.01 | 0.00 | 0.01 | 0.02 |
| coccosphere | 19.82 | 18.15 | 18.15 | 16.78 |
| dev. Std | 0.11 | 0.07 | 0.49 | 0.69 |
| cell | 15.65 | 11.76 | 12.91 | 12.81 |
| dev. Std | 1.39 | 0.29 | 0.82 | 0.91 |
| DSL | 13.10 | 10.77 | 11.78 | 10.12 |
| d.StdL | 1.23 | 1.06 | 0.96 | 0.66 |
| DSW | 11.43 | 9.11 | 10.10 | 8.61 |
| d.StdW | 1.16 | 0.99 | 0.96 | 0.55 |
| ellipticity | 1.15 | 1.19 | 1.17 | 1.18 |

**P. carterae**

|  | Control | OA | CS1 | CS2 |
|---|---|---|---|---|
| pH | 8.03 | 7.82 | 7.88 | 7.61 |
| TA | 2351 | 2313 | 3722 | 4984 |
| fCO2 | 409 | 697 | 968 | 2782 |
| HCO3- | 1946 | 2027 | 3271 | 4700 |
| Ca out | 3.83 | 2.46 | 2.46 | 4.58 |
| μ | 0.52 | 0.53 | 0.51 | 0.52 |
| dev. Std | 0.02 | 0.02 | 0.02 | 0.00 |
| coccosphere | 11.70 | 11.39 | 11.52 | 11.56 |
| dev. Std | 0.10 | 0.32 | 0.09 | 0.44 |
| cell | 9.03 | 9.16 | 9.35 | 9.77 |
| dev. Std | 0.32 | 1.11 | 0.59 | 0.08 |
| DSL | 1.89 | 1.92 | 1.84 | 1.90 |
| d.StdL | 0.14 | 0.16 | 0.13 | 0.10 |
| DSW | 1.17 | 1.22 | 1.22 | 1.19 |
| d.StdW | 0.10 | 0.10 | 0.10 | 0.08 |
| ellipticity | 1.63 | 1.58 | 1.52 | 1.60 |

**Table 6 Carbonate chemistry experiment data. Data presented are the average of three replicates. Growth rate (μ, cell d$^{-1}$); coccosphere, cell diameters (μm); coccolith morphometric analysis were performed on 50 specimens for every treatment and for every replicate. Data represent the average of three replicates: average of coccolith distal shield length (DSL; μm) and coccolith distal shield width (DSW; μm); ellipticity (L/W) diameter; average *E. huxleyi* distal shield elements number (SE) and average distal shield elements width (μm, SEW); average *E. huxleyi* inner tube thickness (μm, tube thick.), *G. oceanica* bridge angle (angle °). Carbon chemistry speciation calculated as the mean of start and end values of measured pH and TA are given.**