# Peer review of "Can morphological features of coccolithophores serve as a reliable proxy to reconstruct environmental conditions of the past?"

_Climate of the Past, 2019_

## Short Comment (SC1) · 22 Oct 2019

Reading quickly the submitted manuscript, I was looking at Figure 3, the hearth of the study (along with Fig.4). However, I don't understand what represent the whiskers around what I suppose are mean. And I found that there were 50 coccoliths measured per treatment only by reading the Tables captions. It should be in the methods. Any stats depends on n so it should be clear and easily accessible. I recommend to show 95%CI (2 sigma on each side) in Figure 3 or switch for boxplot. You should also show 95%CI in Figure 4. Because those are counted coccoliths, you may be able to apply counting uncertainty following Suchéras-Marx et al. 2019 and the "exact" Clopper-

[Figure]

Pearson method easily accessible with PAST 3.06 and updated. The Clopper-Pearson is a good solution because you have in Fig. 4 a lot of numbers close to 0%/100% borders. If you apply it, please rather cite Hammer et al., 2001 (Hammer, Ø., Harper, D.A.T., Ryan, P.D., 2001. PAST: paleontological statistics software package for education and data analysis. Palaeontologia Electronica 4, 1-9.) than Suchéras-Marx et al., 2019. Some words are too small in Fig. 3 and in Tables (d std). It should be the same size everywhere. Good luck Sincerely

———————————————

---

## Short Comment (SC2) · 22 Oct 2019

Caption Figure 4: "any malformations were observed under the different light intensities tested; therefore, percentage of malformations are not represented for these experiments for this species. Furthermore, any malformation was observed for P. carterae and percentage are not shown." What do you mean ? No malformation? That's not clear. Could you rephrase.

———————————————————

---

## Referee Comment (RC1) · Anonymous Referee #1 · 7 Nov 2019

Dear editor, I have reviewed the paper entitled with 'Can morphological features of coccolithophores serve as a reliable proxy to reconstruct environmental conditions of the past?' submitted by Faucher et al. to CP.

In this study, the authors designed different experiment to test the 4 species of coccolithophores' response to the environment settings, such as seawater Mg/Ca, seawater carbonic chemistry, light and nutrient. They found that cell and coccolith sizes behave differently to environment setting among species, while the malformations of coccolith shared a common response to the seawater carbonic chemistry. They concluded that the size or other morphological parameters can not serve as paleoenvironment

proxies, while the higher coccolith malformation rate can be an evidence for high CO2 environment.

In general, the dataset is valuable for both modern and paleo coccolithophore research. I think what they have presented is clear enough except a few details should be added into the Methods part. The main problem is that they did not dig their data enough which leads their paper looks more like a data report rather than a mature article. I think the first author has a lot of experience on the nannofossil evolutions and the other two authors are experts in modern coccolithophore. They could offer a much better discussion for their valuable data. Here are some detailed comments:

1) I suggest that the authors should explain more about why they design these tests on coccolithophore in the Introduction or Methods part, such as the carbonic chemistry experiments are for ocean acidification and Mg/Ca experiments are for the long-time evolution of seawater. How about considering move the first paragraph in discussion into introduction? Because I feel that I am clearer for the study motivation after reading it.

2) I doubt about can the timing of measurement influence the coccolith length results. Because a previous work shows the growth phase can influence the coccolith and cell size (Sheward et al., 2017). That also means if the cells are harvested right after and before dividing, the results should be different. Since there were light-dark cycles in your experiment setting, the authors should mention when they harvest the cells for size measurement and are these timings same among different experiment.

3) I did not find how many coccoliths did they measured in the Methods part, which is important for statistical analyses.

4) In Figure 2, does the 'C' mean 'control experiments'. If so, why the results in different control experiment are quite different? Are the experiment settings in different control experiments different or the same? If the settings are same, why the results so different?

5) I did not find where they discuss the results of ellipticity, rays, bridge angles. That is why I say they should dig their data deeper.

6) They found that different coccolithophores have different response to the environment settings. From my point of view, that is not something surprises me: they should not behave in a similar way. What we really want to know is why is that, so the authors should try to offer an explanation rather than leaving the conclusion in this way. In this case, I do not quite agree with the opinion that coccolith morphological data can only serve as a paleo-proxy when different species have a same response to the environment variations. Well, that is only my idea, and the authors do not have to change their conclusion if they can offer a better discussion than this version.

7) These two papers may help for a rich discussion, Sheward et al. (2017) and Aloisi (2015), which are absent from their references list.

References:

Sheward R M, Poulton A J, Gibbs S J, et al. Physiology regulates the relationship between coccosphere geometry and growth phase in coccolithophores[J]. Biogeosciences, 2017, 14(6): 1493-1509.

Aloisi G. Covariation of metabolic rates and cell size in coccolithophores[J]. 2015.

---

## Referee Comment (RC2) · Mariem Saavedra-Pellitero (Referee) · 15 Nov 2019

GENERAL COMMENTS

In this manuscript, Faucher et al. report how they cultured four extant coccolithophore taxa (Emiliania huxleyi, Gephyrocapsa oceanica, Coccolithus braarudii, and Pleurochrysis carterae) with different environmental variables (light intensity, Mg/Ca ratio, nutrient availability, temperature and carbonate chemistry) and how they assessed the potential response analysing coccolith morphology. Based on the culture experiments, the authors extrapolate to the past and conclude that coccolith morphology cannot be used as a proxy for reconstructions of palaeoenvironmental conditions, but instead

they suggest that malformations can be a good indicator for high CO2 levels in the sedimentary record.

The manuscript is in general well written and follows a logic structure. The authors provide an interesting dataset, relevant for the scientific community, especially researchers working with phytoplankton communities and/or coccolithophores/calcareous nanno-fossils in palaeoclimate and paleoceanographic reconstructions.

The methodology chosen is adequate and the authors explain it shortly but concisely. Still, I think that sometimes the authors could elaborate more (see specific comments and supplementary material, where there are sections with one single sentence).

Faucher et al. generated a valuable high quality dataset, however, they somehow fail to discuss their findings in detail. The discussion part is one of the weakest points of the manuscript at the moment, not only regarding the coccolith morphometries, but also because the authors do not compare their findings/results as much as they could to previously published papers (mostly dealing with a) high CO2 past time intervals, and b) coccolith malformations).

Although I think that this manuscript it is not yet ready for publication and it will need a thorough revision, I still recommend it for publication Climate of the Past. I am confident that the authors will be able to improve this version of the manuscript with the specific comments and suggestions provided.

SPECIFIC COMMENTS:

Abstract: Page (P) 1, lines (L) 15-16. Rephrase; "evolutionarily distinct for millions of years" is misleading and not the case for E. huxleyi and Gephyrocapsa. P 1, L 18-20. I understand that this is one of the main motivations of this piece of research, but the authors are jumping here from hours/days to Millions of years. They need to be more careful linking present-day changes to past evolution. P 1, L 22-25. This conclusion needs to be discussed in more detail (not here, but in section 4).

1. Introduction: P 1, L 27. Add reference after "producers on Earth". P 1, L 28-29. "...exoskeleton (coccosphere) composed of single platelets called coccoliths and nannoliths". Rephrase, being more accurate. If I am not mistaken, some nannoliths can be internal (e.g., Ceratholithus). P 1, L 29 "...ability to precipitate calcium carbonate" = to calcify? P 2, L 34-39. Reword, because this part is a bit confusing for the reader and it sounds contradictory to sentence 39-40. P 2, L 41-43. "In case that coccolith morphology responses to a changing environmental driver are similar in the four species this could be indicative of a response pattern that was conserved over geological timescales". This is a general assumption that the authors do for all the 4 taxa. Can the authors add some reference(s) backing up this? (I.e. explaining why these taxa would have to behave in the same way, I could imagine that would be the case for E. huxleyi and Gephyrocapsa, but it might be worthy to additional information in that regard). P 2, L 47-48. They way this sentence is written is tricky/misleading, because E. huxleyi and Gephyrocapsa are already half of the taxa considered. Also, despite the long evolutionary history mentioned, no traces of the delicate coccoliths of the calcifying species within the family Pleurochrysidaceae have been observed in the fossil record (e.g. De Vargas et al., 2007). Therefore Faucher et al should be careful how they objectively show the state of the art information). P 2, L 47. Double check if Pleurochrysis carterae is the more adequate name for this taxa or if it should be changed to Chrysotila carterae. http://www.mikrotax.org/system/index.php?dir=Coccolithophores/Coccolithales/Pleurochrysidaceae/Chrysotila P 2, L 50-51. Change one of the words "whether" to avoid repetition. 2. Material and methods P 2, L 54. Delete "generally" P 2, L 56. Do you mean nutrient limitations or content? If the authors are referring to limitation, maybe it would be worthy to add a bit more information regarding this. P 2, L 56. Carbonate chemistry...parameters? Maybe it would be worthy to list them here, at the very beginning of the section. P 2, L 57. Which E. huxleyi morphoptype? Do the authors have information about it? There might be different responses by different morphotypes. P 2, L 57-59. It might be worthy to specify where all those strains originally come from. P 3, L 59. Delete ASW,

because it is the only time authors use this abbreviation P 3, L 70. Change "start" for "beginning". P 3, L 71. "It was assured" sounds odd to the reader. P 4, L 106 and 113. Why there are two different references for the program CO2SYS (Lewis and Wallace, 1998 and Schulz et al., 2017)? Why the authors did not use Pierrot et al. (2006) instead of Lewis and Wallace (1998) that seems to be a more recent program? P 4, L 107. Roy et al. (1993) Parenthesis are missing. P 4, L 109. Was this following Gattuso et al. (2010)? Rephrase. P 5, L 124. I sentence should not start in that way. Rephrase or write "Fifteen to ten ml…" P 5, L 130-131. What those references (Langer et al., 2006; Langer et al., 2010) refer to? The way that the malformations were quantified? P 5, L 138. Why a non-linear regression? Elaborate.

3. Results: P 5, L 142. Delete "all", add "selected" after species and add "variable" before light intensity. P 5, L 144. No need for a new paragraph after only one sentence. P 5, L 145. Here and elsewhere in the manuscript. A sentence should not start with an abbreviation or a number, therefore "G. oceanica" should be "Gephyrocapsa oceanica" here. Please, double check this throughout the manuscript (and supplementary material). There are plenty of abreviations at the beginning of sentences, especially in section 3 (L. 153, 155, 156, 160, 161, 162, 163…). P 6, L 166. Inner tube? P 6, L 169. Replace "any" by "no". P 6, L 173. No need for a new paragraph after only one sentence. P 6, L 173. Replace "under" by "at". P 6, L 173. What do the authors mean by rays? T-elements? Revise this and use the adequate nomenclature. P 6, L 176. I suggest to change the title of this section to "Carbonate chemistry parameters" or something along those lines. P 7, L 183. A dot is missing after (Table 5). P 7, L 185. "oceanica formed a high number of malformed coccoliths" sounds odd. Reword if possible. P 7, L 187. Do the authors mean Figure 2 or Figure 4? P 7, L 191. Add "variations" after "carbonate chemistry"

4. Discussion: P 7, L 193. How do we know it is a biological innovation? Any reference for that? P 7, L 194-195. "A great diversification in morphologies occurred in the Mesozoic and Paleocene where many new morphologies occurred." Rephrase. It sounds

like circular reasoning. P 7, L 198. Coccolithophore algae (without s). P 7, L 201-202 "The cause of this impressive number of structures is unknown but there might be a reason connected to the function of coccoliths for the different species to produce such different shapes." This sounds very vague. P 7, L 202-203. I miss a reference here.

P 7-8, L 210-212. "However, fossils and living coccolithophores diverged a long time ago, have a different genetic background and therefore, calcareous nannoplankton in the past and nowadays did and do not necessarily act in the same way to external stress". This sentence kind of undermines some of the author's basic assumption(s) for this study, but I like that they mention this kind information. I even suggest to discuss this more in detail. Maybe it would be worthy also to add something along those lines in the introduction/state of the art (this applies to information provided from L210 to L220) to outline better the main goals, but also limitations, of this research work.

P 8, L235-242. This is one of the main conclusions drawn from this research, but somehow the authors only mention it superficially, and do not even discuss it. Discussion needs to be added, especially considering existing literature (already cited by the authors!) regarding relatively well preserved nannofossils in high pCO2 past environments; e.g., during Paleocene-Eocene Thermal Maximum, O'Dea et al., 2015, Gibbs et al., 2016...). In my opinion this part needs to be improved,

The fact that coccolithohophores are generating malformed coccoliths in extreme/"harsh" boundary conditions (high CO2 concentrations in this very specific case) is something that has been previously noted or at least mentioned by other authors. Still Faucher et al. miss the chance to discuss it in section 4. I recommend the authors to add some more references to make stronger the discussion.

P 8, L243. Change "excess" to "high concentration"

As a general comment, I got the impression that there is a lot of morphometric data displayed in tables and mentioned in the results part, but I miss some discussion on top of L 221-225 comparing to other published studies.

5. Conclusions: P 9, L253. Conclusions P 9, L255. I would rewrite this sentence as: "…in response to temperature, light, nutrient, and Mg/Ca variations". P 9, L260-262. Rephrase this sentence. Make it simpler or split into two sentences.

6. References: Check the way dois are cited, e.g., L271 vs L274.

7. Figures: Figure 1: Is it possible to use italics for the name of the species? What is the meaning of the numbers at the nodes (47, 57…)? Maybe it is worthy to specify what they mean in the caption

L 360. Caption: I suggest to reword it as follows: "Phylogeny and divergence times of the Haptophytes, modified from Liu et al., (2010). Time is indicated in billion years. The species selected for this study are shown in red."

Figure 2. Caption: Based on the (very nice) figure itself, I would change this sentence to "…incomplete and incomplete / malformed for E. huxleyi…"

Figure 3. Is it possible to add the chemistry parameters (i.e., pH, TA…)?. I would also recommend to use the same (or similar) size font. The authors can always use abbreviations if the whole word does not fit.

L 369. Figure 4. Caption and figure. Use incomplete / malformed, as in figure 2.

Table 1. Use the same font size (This applies to all the tables). L 390. Caption (here and elsewhere): double check that these units are correct: "Growth rate ($\mu$)". L 392. Here and elsewhere: "E. huxleyi ray number (rays)" Do you mean T-elements?

I would also recommend the authors to re-structure all the tables. Do not mix different parameters in the same column (e.g., rays and tube thick. In Table 1). I think it would be better to leave gaps or specify that there is no data (as in Table 5) rather than mixing different measurements. It is confusing for the reader.

---

## Author Comment (AC1) · 11 Dec 2019

We thank Dr. Suchéras-Marx for his suggestions and for whishing us the success of our paper.

We wrote in the paper more clearly and in the caption of Fig.3 that we measured 50 coccoliths for every treatment, for every replicate, in every species. We also followed his good advice and we modified Fig.3 from scatter graphs to box plots. We also changed the caption in Fig.4, the font size in Fig.3 and in Tables.

We didn't apply the statistical analyses that he suggested but we accurately looked at

[Figure]

the papers that Dr. Suchéras-Marx suggested to us and we thank him for his advice.

[Figure]

[Figure]

Fig. 3 Box plots of coccolith length from the different experiments. In A) *E. huxleyi*; B) *G. oceanica*; C) *C. braarudii*; D) *P. carterae*. C= control treatment. Light: experiment with 12 different light intensities from 50 to 600 μm photons m⁻² s⁻¹; Ca: calcium manipulation experiment, in 25, [Ca²⁺] =25 mmol L⁻¹; in 50, [Ca²⁺]= 50 mmol L⁻¹; N: nutrient limitation experiment, NL= nitrogen limited conditions; PL= phosphate limited experiment. T: temperature experiment, 22.5 = temperature tested; CC: carbonate chemistry experiment; theoretical CO₂ values; C= 400 ppm, OA= 1000 ppm, CS1= 1000 ppm, CS2= 3000 ppm (for further information see paragraph 3.5). The tops and bottoms of each "box" are the 25th and 75th percentiles of the samples respectively. The red line in the middle of each box is the median. The whiskers, extending above and below each box, represent the furthest observations. Observations beyond the whisker length are marked as outliers (red cross). For the light experiment, 50 specimens were considered for every treatment. For Mg/Ca experiment (Ca), nutrient experiment (N), temperature (T) and carbonate chemistry manipulations (CC) experiments, every box plot represents 150 measurements in total, (50 measurements in each replicate).

**Fig. 1.**

---

## Author Comment (AC2) · 16 Dec 2019

Dear Mariem Saavedra-Pellittero, thank you for the constructive review that was very helpful to improve our manuscript.

Below we respond to all points raised (see original comment by Dr. Mariem Saavedra-Pellittero for details). Note: the pages and lines are from the original file. Please find our point-by-point reply in the pdf file too.

Kind regards Giulia Faucher

[Figure]

Abstract

-L 15-16: Rephrase; "evolutionarily distinct for millions of years" is misleading and not the case for E. huxleyi and Gephyrocapsa".

We rephrased as suggested: "to investigate this, we cultured four living coccolithophore species (Emiliania huxleyi, Gephyrocapsa oceanica, Coccolithus pelagicus subsp. braarudii, and Pleurochrysis carterae) that have been evolutionarily distinct for hundred thousand to millions of years..."

-L 18-20 and L 41-43 (P2) and 210-212 (P-8): I understand that this is one of the main motivations of this piece of research, but the authors are jumping here from hours/days to Millions of years. They need to be more careful linking present-day changes to past evolution.

The idea behind this approach was to use species that diverged with each other many million years ago. If these species would have responded in the same way to one of the parameters that we tested, there could be a good chance that this physiological response was genetically conserved through time. This assumption comes from the idea that the chosen species share common ancestors but followed separate diversification processes and transitions for millions of years (Liu et al., 2010). Furthermore, fossil and living organisms have millions of years in between and therefore, it's true that we are comparing different time scale. But there are millions of years of divergence time among the tested living coccolithophore too and among living and fossil species. To this end, from our point of view, it is possible with caution, to make a comparison between fossil species and living algae. Finally, it's true that E. huxleyi and G. oceanica diverged with each other kyrs ago. We decided, however, to use both species to check whether the response among these two taxa would have been more similar under the tested environmental parameters and if this could be due to their closer genetic relatedness.

We changed the text: "... that have been evolutionarily distinct for hundred thousand to millions of years..."

-L 22-25. This conclusion needs to be discussed in more detail (not here, but in section 4). We thank Dr. Mariem Saavedra- Pellitero for this comment that gave us the chance to dig deeper into malformations in the fossil record. We added a new paragraph in the discussion. See below

Introduction

-L27 Add reference after "producers on Earth.

Reference added: Tyrell and Young, 2010. Tyrrell, T. and Young, J. R.: Coccolithophores, in: Encyclopedia of Ocean Sciences, edited by: Steele, J. H., Turekian, K. K., and Thorpe, S. A., Academic Press, San Diego, 3568–3576, https://doi.org/10.1016/B978-012374473-9.00662-7, 2009

-L28-29: "...exoskeleton (coccosphere) composed of single platelets called coccoliths and nannoliths". Rephrase, being more accurate. If I am not mistaken, some nannoliths can be internal (e.g., Ceratholithus).

We checked and rephrased accordingly. Yes, Ceratholithus is characterized by both nannolith (ceratholith) inside but also outside the cell

Cros, L., Kleijne, A., Zeltner, A., Billard, C., & Young, J. R., 2000. New examples of holococcolith–heterococcolith combination coccospheres and their implications for coccolithophorid biology. Marine Micropaleontology, 39(1-4), 1-34.

-L29 "...ability to precipitate calcium carbonate" = to calcify?

Rephrased accordingly and changed as suggested by the referee.

-L34-39 Reword, because this part is a bit confusing for the reader and it sounds contradictory to sentence 39-40.

We added a sentence that could help to understand the aim of our work and the chose

approach.

"The primary goal of our study was to understand if physiological experiments with contemporary species are a valid tool to reconstruct responses of ancient coccolithophores to environmental change in the geological record".

-L41-43. In case that coccolith morphology responses to a changing environmental driver are similar in the four species this could be indicative of a response pattern that was conserved over geological timescales". This is a general assumption that the authors do for all the 4 taxa. Can the authors add some reference(s) backing up this? (I.e. explaining why these taxa would have to behave in the same way, I could imagine that would be the case for E. huxleyi and Gephyrocapsa, but it might be worthy to additional information in that regard).

We modified the text to clarify our approach and we replied to this comment in the line above.

"The primary goal of our study was to understand if physiological experiments with contemporary species are a valid tool to reconstruct responses of ancient coccolithophores to environmental change in the geological record. To test this assumption, we did a series of identical stress test experiments with four selected modern species that have been evolutionarily distinct since hundred thousand to millions of years (Fig.1)."

-L 47-48. They way this sentence is written is tricky/misleading, because E. huxleyi and Gephyrocapsa are already half of the taxa considered. Also, despite the long evolutionary history mentioned, no traces of the delicate coccoliths of the calcifying species within the family Pleurochrysidaceae have been observed in the fossil record (e.g. De Vargas et al., 2007). Therefore Faucher et al should be careful how they objectively show the state of the art information).

We changed the sentence to be clearer on the hypothesis of our experiments.

"Our hypothesis was: in case that coccolith morphology responses to a changing environmental driver are similar in the four species this could be indicative of a response pattern that was conserved over geological timescales. In other words, if species conserve a similar response to certain types of environmental change for geological timescales despite very different evolutionary trajectories, then this would strengthen our confidence that responses recorded for modern species also apply for the geological past. The assumption that this approach is valid has often made but, to the best of our knowledge, not been further tested so far."

-L47 Double check if Pleurochrysis carterae is the more adequate name for this taxa or if it should be changed to Chrysotila carterae. http://www.mikrotax.org/system/index.php?dir=Coccolithophores/Coccolithales/Pleurochrysidaceae/Chrysotila

Chrysotila carterae and Pleurochrysis carterae are synonyms. We left the name Pleurochrysis carterae since this is the name labeled in the stock culture and it was already used for previous work (see Faucher et al., 2017b). http://www.mikrotax.org/Nannotax3/index.php?id=290.

-L 50-51. Change one of the words "whether" to avoid repetition. Changed

Material and Methods

-L54 Delete "generally modified Modified

-L56 Do you mean nutrient limitations or content? If the authors are referring to limitation, maybe it would be worthy to add a bit more information regarding this.

We mean nutrient limitation that was added to the text. More information about this experiment is written in paragraph 2.1.3. Therefore, we didn't change this part.

-L56 Carbonate chemistry: : :parameters?

Carbonate chemistry: four different conditions were tested and we listed the information in the paragraph with the specifics about this experiment 2.1.5. Therefore, we didn't change this part.

-L57 Which E. huxleyi morphoptype? Do the authors have information about it? There might be different responses by different morphotypes. L 57-59. It might be worthy to specify where all those strains originally come from.

Information added in the text: "Emiliania huxleyi (strain RCC 1216, pelagic, from the Tasmanian sea), Gephyrocapsa oceanica (strain RCC 1303, pelagic, from the France coast of the Atlantic Ocean), Coccolithus pelagicus subsp. braarudii (strain PLY182G, it will be called hereafter C. braarudii, pelagic, from the English channel, Atlantic Ocean), and Pleurochrysis carterae (unknown strain number, coastal species)"

-L59 Delete ASW, ASW deleted.

-L70 Change "start" for "beginning". Changed accordingly.

-L71 "It was assured" sounds odd to the reader Changed and deleted.

-L106-113 Why there are two different references for the program CO2SYS (Lewis and Wallace, 1998) Schulz et al., 2017)? Why the authors did not use Pierrot et al. (2006) instead of Lewis and Wallace (1998) that seems to be a more recent program?

We checked and the right reference was added to the text which is, Pierrot et al., (2006). In line 113, Schulz et al., (2007) is referred to the recalculation of pH.

-L107 parentheses added. -L109 rephrased

-L124 I sentence should not start in that way. Rephrase or write "Fifteen to ten ml: : :" P

The sentence was rephrased.

-L130-131 What those references (Langer et al., 2006; Langer et al., 2010) refer to?

The way that the malformations were quantified?

It was added to the text that we followed Langer et al., (2006 and 2010) to group normal, malformed, incomplete and incomplete/malformed coccoliths.

-L138. Why a non-linear regression? Elaborate.

We used the nonlinear regression because for the light experiment we decided to perform an experiment with a high number of treatment levels but at the expense of replication (one replicate per sample).

Results

-L142 Delete "all", add "selected" after species and add "variable" before light intensity.

Changed accordingly.

-L144 No need for a new paragraph after only one sentence.

Paragraph deleted.

-L145 Here and elsewhere in the manuscript. A sentence should not start with an abbreviation or a number, therefore "G. oceanica" should be "Gephyrocapsa oceanica" here. Please, double check this throughout the manuscript (and supplementary material). There are plenty of abreviations at the beginning of sentences, especially in section 3 (L. 153, 155, 156, 160, 161, 162, 163: : :).

Abbreviation checked and modified through the text.

-L166 Inner tube?

Inner tube, changed in the text.

-L169 Replace "any" by "no". L 173. No need for a new paragraph after only one sentence. L 173. Replace "under" by "at".

Modified as requested.

-L173 What do the authors mean by rays? T-elements? Revise this and use the adequate nomenclature.

Rays are now called distal shield elements.

-L176 I suggest to change the title of this section to "Carbonate chemistry parameters" or something along those lines. L 183. A dot is missing after (Table 5).

Title modified accordingly and dot added.

-L185 "oceanica formed a high number of malformed coccoliths" sounds odd. Reword if possible.

Text modified.

-L187 Do the authors mean Figure 2 or Figure 4? It refers to Fig. 2. The text was changed accordingly

-L191: add "variations" after "carbonate chemistry" Text modified.

Discussion

-L193: How do we know it is a biological innovation? Any reference for that? A reference was added and text modified.

De Vargas, C., Aubry, M. P., Probert, I. A. N., Young, J.: Origin and evolution of coccolithophores: from coastal hunters to oceanic farmers, In Evolution of primary producers in the sea (pp. 251-285) Academic Press, 2007.

-L194-195 "A great diversification in morphologies occurred in the Mesozoic and Paleocene where many new morphologies occurred." Rephrase. It sounds like circular reasoning.

Rephrased

-L 198. Coccolithophore algae (without s) s deleted

-L 201-202 "The cause of this impressive number of structures is unknown but there might be a reason connected to the function of coccoliths for the different species to produce such different shapes." This sounds very vague.

We modified the sentence in order to be less vague and better express our statement.

"The cause of this impressive number of structures is unknown but there might be a reason connected to the function of coccoliths for the different species to produce such different shapes ranging from protection against excess sun light and/or against grazing (Monteiro et al., 2016)."

-L202: I miss a reference here. Reference added

-L210-212. "However, fossils and living coccolithophores diverged a long time ago, have a different genetic background and therefore, calcareous nannoplankton in the past and nowadays did and do not necessarily act in the same way to external stress". This sentence kind of undermines some of the author's basic assumption(s) for this study, but I like that they mention this kind information. I even suggest to discuss this more in detail. Maybe it would be worthy also to add something along those lines in the introduction/state of the art (this applies to information provided from L210 to L220) to outline better the main goals, but also limitations, of this research work.

We replied to this referee comment, above.

-L235-242. This is one of the main conclusions drawn from this research, but somehow the authors only mention it superficially, and do not even discuss it. Discussion needs to be added, especially considering existing literature (already cited by the authors!) regarding relatively well preserved nannofossils in high pCO2 past environments; e.g.,

during Paleocene-Eocene Thermal Maximum, O'Dea et al., 2015, Gibbs et al., 2016: : :). In my opinion this part needs to be improved. The fact that coccolithophores are generating malformed coccoliths in extreme/" harsh" boundary conditions (high $CO_2$ concentrations in this very specific case) is something that has been previously noted or at least mentioned by other authors. Still Faucher et al. miss the chance to discuss it in section 4. I recommend the authors to add some more references to make stronger the discussion.

We thank Dr. Saavedra-Pellittero for this comment and for the opportunity she gave us to dig into malformation in the fossil record. We added a paragraph to the text.

"In the fossil record there are several examples of intervals characterized by high abundances of malformed specimens, linked to the low calcite saturation state of the ocean (Jiang and Wise, 2006; Raffi and De Bernardi, 2008; Agnini et al., 2007; Erba et al., 2010; Bralower and Self Trail, 2016). Different authors argued for high $pCO_2$ influence on causing these malformations during the Mesozoic OAEs, Paleocene-Eocene Thermal Maximum (PETM) and Eocene Thermal Maximum 2. All these intervals were characterized by excess $CO_2$ concentrations and/or slightly reduced pH. Malformations were expressed in different ways: it was represented by variation in ellipticity of coccoliths (Erba et al., 2010), asymmetry (Agnini et al, 2007), irregular arrangement and length of their rays and diminished calcification in some nannoliths, (Jiang and Wise, 2006; Mutterlose et al., 2007; Raffi and De Bernardi, 2008; Bralower and Self Trail, 2016). The short stratigraphic ranges where these malformations occurred, during the core of ocean perturbations, indicated that pH played a role in inducing the production of these aberrant specimens (Mutterlose et al., 2007; Erba et al., 2010). There is still not a clear explanation of the reason why only some species of calcareous nannoplankton were producing aberrant specimens, and there is not a general consensus on the role of carbonate chemistry on coccolithophore biomineralization (Gibbs et al., 2010; Gibbs et al, 2016). However, a more recent work, provides a plausible explanation of what might have happened during the PETM, where only some species moved

and inhabited the deep part of the photic zone, to possibly refuge from stressful warm and eutrophic conditions of the surface water, but had to deal with lower saturation conditions that induced the detected malformations for these taxa (Bralower and Self Trail, 2016). The increase in the percentage of malformed coccoliths observed in our experiments, suggests a more universal occurrence of malformation in modern coccolithophore species under low pH."

References:

Agnini, C., Fornaciari, E., Rio, D., Tateo, F., Backman, J., Giusberti, L.: Responses of calcareous nannofossil assemblages, mineralogy and geochemistry to the environmental perturbations across the Paleocene/Eocene boundary in the Venetian Pre-Alps. Marine Micropaleontology, 63(1-2), 19-38, https://doi.org/10.1016/j.marmicro.2006.10.002, 2007.

Bralower, T. J., Self‐Trail, J. M.: Nannoplankton malformation during the Paleocene‐Eocene Thermal Maximum and its paleoecological and paleoceanographic significance. Paleoceanography, 31(10), 1423-1439, https://doi.org/10.1002/2016PA002980, 2016

Gibbs, S. J., Stoll, H. M., Bown, P. R., Bralower, T. J.: Ocean acidification and surface water carbonate production across the Paleocene–Eocene thermal maximum. Earth and Planetary Science Letters, 295(3-4), 583-592, https://doi.org/10.1016/j.epsl.2010.04.044, 2010.

Gibbs, S. J., Bown, P. R., Ridgwell, A., Young, J. R., Poulton, A. J., O'Dea, S. A.: Ocean warming, not acidification, controlled coccolithophore response during past greenhouse climate change, Geology, 44(1), 59-62, https://doi.org/10.1130/G37273.1, 2016.

Jiang, S., Wise Jr, S. W.: Surface-water chemistry and fertility variations in the tropical Atlantic across the Paleocene/Eocene Thermal Maximum as evidenced by calcareous

nannoplankton from ODP Leg 207, Hole 1259B. Revue de micropaléontologie, 49(4), 227-244, https://doi.org/10.1016/j.revmic.2006.10.002, 2006.

Mutterlose, J., Linnert, C., Norris, R.: Calcareous nannofossils from the Paleocene–Eocene Thermal Maximum of the equatorial Atlantic (ODP Site 1260B): evidence for tropical warming. Marine Micropaleontology, 65(1-2), 13-31, https://doi.org/10.1016/j.marmicro.2007.05.004, 2007.

Raffi, I., De Bernardi, B.: Response of calcareous nannofossils to the Paleocene–Eocene Thermal Maximum: Observations on composition, preservation and calcification in sediments from ODP Site 1263 (Walvis Ridge—SW Atlantic). Marine Micropaleontology, 69(2), 119-138, https://doi.org/10.1016/j.marmicro.2008.07.002, 2008.

-L253: Change "excess" to "high concentration" Modified accordingly

Conclusion

-L255: I would rewrite this sentence as: ": : :in response to temperature, light, nutrient, and Mg/Ca variations". 260-262. Rephrase this sentence. Make it simpler or split into two sentences.

L255 Sentence was rephrased L260-262 The sentence was rephrased.

6. References:

Check the way dois are cited, e.g., L271 vs L274.

Checked and modified as requested.

Figures and tables

-Figure 1: Is it possible to use italics for the name of the species? What is the meaning of the numbers at the nodes (47, 57: : :)? Maybe it is worthy to specify what they mean in the caption L 360. Caption: I suggest to reword it as follows: "Phylogeny and

divergence times of the Haptophytes, modified from Liu et al., (2010). Time is indicated in billion years. The species selected for this study are shown in red."

Caption and Fig.1 modified accordingly.

Caption: "Figure 1: Phylogeny and divergence times of the Haptophytes, modified from Liu et al., (2010). Time is indicated in billion years. The species selected for this study are shown in red. Nodes, representing following divergence, used for calibrating the tree with dates from the fossil record, are shown in green."

-Figure 2. Caption: Based on the (very nice) figure itself, I would change this sentence to ": : :incomplete and incomplete / malformed for E. huxleyi: : :"

Modified accordingly

-Figure 3 Is it possible to add the chemistry parameters (i.e., pH, TA: : :)?. I would also recommend to use the same (or similar) size font. The authors can always use abbreviations if the whole word does not fit.

Fig. 3 Modified accordingly. The chemistry parameters are now added in the caption.

-Figure 4 Caption and figure. Use incomplete / malformed, as in figure 2.

Fig. 4 Modified accordingly.

-Table 1. Use the same font size (This applies to all the tables). L 390. Caption (here and elsewhere): double check that these units are correct: "Growth rate (_)". L 392. Here and elsewhere: "E. huxleyi ray number (rays)" Do you mean T-elements?

Table 1: Rays were intended to be the distal shield elements. They are now named "distal shield elements" and the abbreviation are SE for the mean number of distal shield elements and SEW for the average width. The font size, growth rate units are changed in all tables.

-I would also recommend the authors to re-structure all the tables. Do not mix different

parameters in the same column (e.g., rays and tube thick. In Table 1). I think it would be better to leave gaps or specify that there is no data (as in Table 5) rather than mixing different measurements. It is confusing for the reader.

We followed the advice for the tables and re-structured them.

Please also note the supplement to this comment:
https://www.clim-past-discuss.net/cp-2019-84/cp-2019-84-AC2-supplement.pdf

[Figure]

[Figure]

Fig. 1.

---

## Author Comment (AC3) · 16 Dec 2019

We thank the anonymous reviewer for the helpful comments on our manuscript. The reviewer also provided several editorial suggestions, which we all considered in our revision of the text. Below we focused our discussion on the more topical points raised by the reviewer.

With regards Giulia Faucher

1) RC: I suggest that the authors should explain more about why they design these tests on coccolithophore in the Introduction or Methods part, such as the carbonic

chemistry experiments are for ocean acidification and Mg/Ca experiments are for the long-time evolution of seawater. How about considering move the first paragraph in discussion into introduction? Because I feel that I am clearer for the study motivation after reading

AC: We thank the referee for the suggestion: we discuss a lot on the introduction and we tweaked it to make the motivation of our work clearer. However, we decided to leave the details about the paleo-record out of this chapter. Indeed, we think that a short introduction is more effective to the reader. We hope that the reviewer agrees on our idea of living the text fully tailored towards the primary goal of our experiments, and aligned the introduction with the title. Finally, the paleo-context is carefully considered in the discussion.

2) RC: I doubt about can the timing of measurement influence the coccolith length results. Because a previous work shows the growth phase can influence the coccolith and cell size (Sheward et al., 2017). That also means if the cells are harvested right after and before dividing, the results should be different. Since there were light-dark cycles in your experiment setting, the authors should mention when they harvest the cells for size measurement and are these timings same among different experiment.

AC: We thank the referee for suggesting us to read and take into consideration the very interesting paper by Sheward et al., (2017). This study demonstrated that the coccosphere, cell size and coccolith number of the analyzed species are influenced by the growth phase that the algae are experiencing. Coccolith sizes don't have any direct link with coccosphere diameters or growth stages. We underline that in our experiments, to avoid an artifact in our data, all measurements were started and conducted at the same time for every experiment that we performed. This information was added to the text.

3) RC:I did not find how many coccoliths did they measured in the Methods part, which is important for statistical analyses.

[Figure]

AC: We measured 50 specimens for every treatment and every replicate. This information was highlighted in the text and in the caption of the figures (see 2.3).

4) RC: In Figure 2, does the 'C' mean 'control experiments'. If so, why the results in different control experiment are quite different? Are the experiment settings in different control experiments different or the same? If the settings are same, why the results so different?

AC: It was added in the caption that C means control conditions. When we performed the experiment, we decided to investigate the same parameter simultaneously for all the tested species. But, the 5 parameters were tested in different experiments performed in a row at different times (different months and years). As indicate by Langer et al., (2012), the morphology of a strain that grew under similar conditions can change over time. The author highlighted, however, that these changes occur at a timescale that is longer of a typical duration of a culture experiment. That means that the experiment validity is not impaired, however, every experiment needs to be considered by itself.

Langer, G., Oetjen, K., & Brenneis, T. (2013). On culture artifacts in coccolith morphology. Helgoland Marine Research, 67(2), 359.

5) RC: I did not find where they discuss the results of ellipticity, rays, bridge angles. That is why I say they should dig their data deeper.

AC: The referee is right. We measured many parameters due to different morphologies of the tested species (e.g. E. huxleyi distal shield element average number, G. oceanica bridge angle). The data are represented in tables for every experiment. However, these morphological features are species-specific and it would have not been possible to compare to metrics for the other coccolithophores. The comparison among different species was the main goal of our study. Therefore, we think that it makes more sense to consider morphological features that can be measured for all coccolithophores (e.g. coccolith size).

6) RC: They found that different coccolithophores have different response to the environment settings. From my point of view, that is not something surprises me: they should not behave in a similar way. What we really want to know is why is that, so the authors should try to offer an explanation rather than leaving the conclusion in this way. In this case, I do not quite agree with the opinion that coccolith morphological data can only serve as a paleo-proxy when different species have a same response to the environment variations. Well, that is only my idea, and the authors do not have to change their conclusion if they can offer a better discussion than this version.

AC: We agree with referee1: it is indeed not surprising that morphology responds differently among species to changing environmental conditions. This was also our hypothesis before we conducted the experiments. However, this assumption (conserved response over geological timescales) is frequently made in studies that investigate responses in living coccolithophores to explain paleo-responses. Therefore, we tested if this frequently made assumptions are justifiable. Furthermore, we were interested if there is something like a "universal response" at least to some of the tested environmental parameters. Such a universal response among evolutionarily very distinct coccolithophores could suggest that the reaction is "conserved" for geological timescales and would improve our confidence that morphology could be used as a paleo-proxy. We only find such a rather universal response for the malformation response to carbonate chemistry. In this regard, we thank the referee for suggesting us to take into consideration the paper by Aloisi, (2015). In this paper, the author suggests that the environment controls coccolith sizes (and or morphology) via a physiological effect and for that reason, he managed to build a model that simulates growth rates and sizes of cells, applicable to the whole coccolithophore community.

7) AC: These two papers may help for a rich discussion, Sheward et al. (2017) and Aloisi (2015), which are absent from their references list.

References:

Sheward R M, Poulton A J, Gibbs S J, et al. Physiology regulates the relationship between coccosphere geometry and growth phase in coccolithophores[J]. Biogeosciences, 2017, 14(6): 1493-1509.

Aloisi G. Covariation of metabolic rates and cell size in coccolithophores[J]. 2015

AC: We thank the referee for the suggestions. The two papers were considered.

---

## Author Response (AR1)

**Editor Decision: Reconsider after major revisions** (08 Jan 2020) by Erin McClymont
Comments to the Author:
Constructive comments have been received by two reviewers and a short online comment. I thank the authors for responding to each of these and considering how to incorporate the suggested changes and clarifications. On the whole the reviewer concerns have been addressed, but there are several where clarification is still required and/or edits to the manuscript should be considered.

**1) Reviewer 1's first comment indicated that there could be greater value in introducing more of the context and rationale for the study in the Introduction. The authors have replied that they prefer to keep a short introduction and leave the detail in the Discussion section. I agree with Reviewer 1 that some further details on the palaeo-context and application of these morphological changes would be valuable, and so ask the authors to consider increasing the detail (either in the Introduction or Discussion, as they see fit): for example, can the authors explain how these previous papers have used morphological changes for palaeoclimate reconstruction, rather than just citing the papers? Or where have debates emerged in the literature about the interpretations of morphological changes? I agree with the Reviewers that at present the manuscript needs some more depth, so that readers can see the link to the important issues in palaeoclimate interpretation. For example, in response to Review 1 comment 6 the authors note that "frequently made assumptions" in the literature are being tested (and see reply to Reviewer 2 lines 47-48), but I did not find these assumptions to be clearly explained nor cited in the manuscript. Reviewer 2 (line 41-43) also suggests citing a source to justify the assumption being tested: if this can't be done (as the author reply seems to suggest: the response does not clarify this), then I agree that a few lines of additional explanation would help the reader to understand this assumption. The authors do a good job where they address Reviewer 2 concerns about the malformations under higher CO2 (response to lines 235-242); this approach could be applied elsewhere. In summary, by addressing these concerns by the reviewers and explaining more about the literature context, I think a stronger foundation for the study will in turn lead to increased clarity about how these results can be applied**.

*We thank the Editor for her recommendations that allow us to reconsider both the introduction and discussion chapters: we confess that we discuss a lot about all issues that the Editor point out and especially about point 1 and 3. Finally, we decided to add some lines in the introduction (L 37-44) to clarify the application of morphological changes in the geological record making a few examples. We also added a sentence in the discussion part (L230-232) to explain the process of associating morphological studies with paleo-environmental conditions.*
*However, we still think that a short introduction and discussion that focused on the main question of our work could make this article more interesting, readable and appealing. The primary aim of our work was to understand if physiological experiments with contemporary species are a valid tool to reconstruct responses of ancient coccolithophores to environmental change in the geological record and this is also the reason for which we didn't dig into species-specific morphological variations.*

*To our knowledge, this work for the first time evidences the weakness of associating nannofossil morphologies and environmental parameters supporting this approach with observations and data from living coccolithophore species. This practice assumes that coccolithophore algae preserve the same mechanism through time towards one (or more) environmental stress/parameter. As far as we know, with this work, for the first time, someone looked deeply into coccolith morphologies under environmental parameters that were stated to induce coccolith variations in the fossil record. Since now, the idea was that both living and fossil species react and reacted in the same way to extreme oceanic conditions given the fact that they all belong to the same group (Haptophyte).*
*We would like to emphasize that one of the authors of this manuscript, applied the same approach in two papers that are cited in the text (Faucher et al, 2017a; Faucher et al., 2017b) and therefore, since the beginning of these experiments, we firstly questioned ourselves the validity of the approach that we also previously used.*

*Having said that, we tried to give more information on the paleo-context we took inspiration from to perform our experiments, in both the introduction and discussion. We wrote a further sub-chapter that was intended to be placed before the (already written) discussion chapter, where we talked about the possible influence of the five tested parameters for coccolith production in the past ocean. However, we still think that this detracts from the main message of our manuscript and we finally decided to delete it. We think that, eventually, this part could be included in the Supplementary file as a "State of the art" paragraph to show the background of our work and underline that we consider most of the literature on the topic of morphometry and environmental condition in the past ocean. We ask the Editor her opinion for this decision hoping that she agrees on our point of view. We leave the written paragraph down here and we also added it in the supplementary file (S1).*

*Geological background - state of the art*
*Several authors tried to link nannofossil morphological variations with detected environmental conditions: the process assumes to analyze nannofossil species through a sedimentary succession and to evaluate the presence of shape or size anomalies in the considered interval of time. The eventual detected morphological variations are then linked to*

*independent paleo-environmental data (e.g. sea surface temperature reconstruction (SST), $CO_2$) to find the environmental driver for the identified morphological variation*

*Indeed, the past oceans were characterized by episodes of anomalous or extreme sea-water conditions that could have possibly influenced the phytoplanktonic communities. A good example is the oscillations between "calcite seas" and "aragonite seas" (Sandberg, 1983) that possibly influenced the productivity of calcareous nannoplankton in different times (Erba, 2006). The amount of massive amount of chalk deposited during the Late Cretaceous is a good illustration of a high productivity time for calcareous nannoplankton probably permitted by a shift in seawater chemistry towards a very high level of Ca. In parallel, rising Mg/Ca ratio during the Cenozoic and up to present days is correlated to a reduction in coccolithophore diversity and coccolith thickness (Bown et al., 2004). Also, locally, light could have played a major role for coccolithophore calcification in the past ocean: it was documented that during episodes of intensified continental weathering, more clastic particles were transported into the sea and in the coastal area might have diminished the depth of the photic zone (Lechler et al., 2015). The reduction in light availability was associated with habitat changes of the photoautotrophic primary producers that produced smaller coccoliths to be able to dwell in shallower depth and compensate for the reduction in sunlight (Lübke and Mutterlose, 2016). The calcareous phytoplanktonic communities in past oceans were also disturbed by intervals with excess $CO_2$ concentrations related to intense volcanic activity. Modifications in size and morphology of calcareous nannofossil during times were $CO_2$ reached up to 1000-2000 ppm, were interpreted as a transient response to survive progressively increasing surface-water acidification (Erba et al., 2010, Lübke et al., 2015; Faucher et al., 2017). Besides, environmental constraints for calcareous nannoplankton growth, involve the ocean trophic level: in the fossil record, some authors linked the decrease in nutrient availability, with reductions of abundances and sizes of some calcareous nannofossil species (Linnert and Mutterlose, 2012). Other authors, on the contrary, detected similar size reductions in several ocean areas characterized by both oligotrophic and mesotrophic seawater conditions (Bornemann et al., 2006; Faucher et al., 2017). Finally, the ocean in its history was subjected to variation in temperature: a strong hydrothermal activity on one hand and an intense continental weathering, on the other hand, were the main triggers of respectively $CO_2$ released and $CO_2$ sequestration, that in turns, often produced a concomitant increase or decrease in SST. Episodes characterized by relatively low SST were sometimes related to small coccolith sizes (Bornemann and Mutterlose, 2006). However, opposite behaviors were also observed (size decreases under extremely warm conditions, Erba et al., 2010; Lübke et al., 2015) in the same species in different geological intervals.*

*References:*

*Bornemann, A. and Mutterlose J.: Size analyses of the coccolith species Biscutum constans and Watznaueria barnesiae from the Late Albian "Niveau Breistroffer" (SE France): taxonomic and palaeoecological implications, Geobios, 39(5), 599-615, https://doi.org/10.1016/j.geobios.2005.05.005, 2006.*

*Bown, P. R., Lees, J. A., Young, J. R.: Calcareous nannoplankton evolution and diversity through time: In Coccolithophores, pp. 481-508. Springer Berlin Heidelberg, 2004.*

*Erba, E.: The first 150 million years history of calcareous nannoplankton: biosphere–geosphere interactions, Palaeogeography, Paleoclimatology, Paleoecology, 232, 2, 237-250, https://doi.org/10.1016/j.palaeo.2005.09.013, 2006.*

*Erba, E., Bottini, C., Weissert, H. J., Keller, C. E.: Calcareous nannoplankton response to surface-water acidification around Oceanic Anoxic Event 1a, Science, 329(5990), 428-432, https://doi.org/10.1126/science.1188886, 2010.*

*Faucher, G., Erba, E., Bottini, C., Gambacorta, G.: Calcareous nannoplankton response to the latest Cenomanian Oceanic Anoxic Event 2 perturbation, Rivista Italiana di Paleontologia e Stratigrafia (Research In Paleontology and Stratigraphy), 123(1), 2017a.*

*Lechler, M., von Strandmann, P. A. P., Jenkyns, H. C., Prosser, G., & Parente, M.: Lithium-isotope evidence for enhanced silicate weathering during OAE 1a (Early Aptian Selli event). Earth and Planetary Science Letters, 432, 210-222. 2015 https://doi.org/10.1016/j.epsl.2015.09.052.*

*Linnert, C., Mutterlose, J.: Biometry of Cenomanian–Turonian placoliths: a proxy for changes of fertility and surface-water temperature? Lethaia, 46(1), 82-97, 2012 https://doi.org/10.1111/j.1502-3931.2012.00323.x.*

*Lübke, N. and Mutterlose, J.: The impact of OAE 1a on marine biota deciphered by size variations of coccoliths, Cretaceous Research, 61, 169-179, https://doi.org/10.1016/j.cretres.2016.01.006, 2016.*

*Lübke, N., Mutterlose, J., Bottini, C.: Size variations of coccoliths in Cretaceous oceans, a result of preservation, genetics and ecology? Marine Micropaleontology, 117, 25-39, https://doi.org/10.1016/j.marmicro.2015.03.002, 2015.*

*Sandberg, P. A.: An oscillating trend in Phanerozoic non-skeletal carbonate mineralogy. Nature, 305(5929), 19-22, 1983.*

**2) Reviewer 1 queried the different control conditions exhibited in Figure 2. It would be useful for the authors to include in their main text or figure caption, the information about the different timing of the experiments to account for the differences. I'm not clear from the reply whether the same control sample was used for all experiments, but stored in between, or if different control samples were generated each time an experiment was started. Could the authors please clarify?**

*Different control samples were generated every time for every experiment from the stock culture. The controls have always the same medium and were treated in the same ways in every experiment. We added in the text the timing of the experiments. See the caption in Fig. 2.*

**3) Reviewer 1 queried why the wider range of morphological measurement parameters was not outlined in the manuscript. Although the authors indicate that in some way this data might detract from their focus of comparing species (and the main assumption they outline for testing), could this information not be valuable for considering the within-species responses, and providing some interesting discussion on what kinds of morphological changes can be observed? I ask the authors to consider whether this information could be included, even though the measurements are not uniform across all species. It could make the manuscript a more comprehensive discussion of morphological change, and add to the finding that different species have different responses.**

*We thank the Editor for her suggestion that allows us to review all the data collected and re-checked all collected morphological parameters.*
*We modified some sentences of the results underlining morphometric information that was already shown in Tab.1.*

*We had a long and deep discussion among us, about the possibility to dig into the species-specific responses for every experiment but at last, we decided to leave the discussion as it is without adding any further paragraph. In the process of writing this manuscript, in the draft version before the submission process, we organized the discussion with six sub-chapter, one for every experiment and a final part where we summarize the observed responses. However, at the end and after spending time and energy in writing those parts, we realized that this structure, the analysis of the available literature for both living and fossil organisms for every experiment, transformed our manuscript into a review paper that wasn't our purpose. We took the drastic decision to delete that part because we thought at that time and we still think that all these paragraphs would have diverged from the key message of our work and would have made little of the answer to our questions.*
*To be more explicit, we entirely agree with the Editor that there are some species-specific changes to the tested parameters and that our dataset has lots of potential in digging more on the response of single coccolithophore species towards environmental stress and we, therefore, understand the idea of the Editor to extend the discussion into within-species responses. However, we believe that analyzing the responses of every species towards every environmental parameter and understand the reason why coccolith react in a certain way towards an environmental condition, is not the goal of this paper.*
*We hope to convince the Editor that a discussion that focuses on the major question we asked ourselves ("Can morphological features of coccolithophores serve as a reliable proxy to reconstruct environmental conditions of the past?) is more desirable.*

**Minor concerns:**
**Reviewer 2 noted confusion about the timing of the divergences. I ask the authors to separately state the timing of the divergence of E.huxleyi and G.oceanica compared to the much older divergence of G.oceanica with the other two species. The timing of E.huxleyi / G.oceanica divergence is never stated, but should be somewhere, especially as it is not visible on Figure 1. Mixing the 'hundred thousand years' with millions of years in the suggested edit doesn't provide clarity.**

*This information was added to the text (line 77).*

**Reviewer 2 also indicated that on Fig 1 the numbers on the nodes were not clearly explained. The authors have added "Node, representing following divergence, used for calibrating...". I'm not still not clear what this numbers mean: is some text missing here?**

*The caption was changed. Fig. 1 is modified from Liu et al., (2010). The nodes represent divergences of species. The numbers represent some of the progressive nodes that Liu et al., (2010) wanted to highlight. For every node the time of divergence was calculated. For example, node 57 corresponds to the divergence between Coccolithus pelagicus and Helicosphaera carteri that was calculated as 220 Ma. For the divergence time of all nodes, see Liu et al., 2010.*

**Reviewer 2 queried (for line 138) why non-linear regression was used. The author reply does not provide clarification: is it because no linear relationship was determined? or because the structure of the changing light intensity experiments does not lend itself to linear regression? or something else?**

*First of all, for the light experiment we preferred to design an experiment with many treatments in the expense of the number of replicates following Cottingham et al., (2005). Therefore, a high number of treatment levels was set up with no replication. We used a non-linear regression because we observed non-linearities in the response variable,*

**The change to Figure 3 is valuable for showing the detailed structure of the data more clearly. Comments have raised concern about how easy the text on the figures is to read. For Figure 3 I would recommend considering plotting this graphic as 2 x 2 panels rather than 1 x 4, which would allow larger versions of each panel to be displayed. The authors should also consider increasing the font size wherever they are able, across Figures 3 and 4.**

*Figure 3 was changed following the Editor advice. The font sizes were increased when possible in Fig. 3 and 4.*

In a revised version of the manuscript the authors should check that the number of coccoliths observed is clearly stated. This was raised several times and although the authors indicate that "this information was highlighted in the text and in the caption" (reply to reviewer 1) I did not find it in the original manuscript either.

*We checked the text. It is now written in: line 142; in the captions in Fig. 3 (lines 481-484), in Fig. 4 (line 505), Tab.1 (line 517), Tab.2 (line 530), Tab. 3 (lines 550), Tab. 4 (line 559) and Tab. 5 (line 589).*

*All the variation in the text are in:*
*Red: referees requests.*
*Underline in grey: Editor's requests.*

*With kind regards*
*On behalf of all co-authors*
*Giulia Faucher*

---

## Author Response (AR2)

Dear Editor,

We thank you and Dr. Mariem Saavedra-Pellitero for the second revision of our manuscript. You carefully read all the manuscript and we thank you for all the suggestions you gave us. We follow your advice and we made most of the changes that you asked.

Primarily, we revised the introduction and we changed the order of the key messages following the order Dr. Mariem Saavedra-Pellitero recommended.

Moreover, we decided to add one new table (Table 1) in the introduction. In this table, we reported all the papers that we could find in the literature, where the morphological and morphometrical variations of calcareous nannoplankton observed in the fossil record, were compared with the results obtained from experiments with living coccolithophores. We, therefore, excluded from this list, papers that showed size and shape analyses of calcareous nannofossils without considering data from culturing experiments (e.g. Erba et al., 2019).

Moreover, in the introduction, we added some references that were missing (lines 33-34).

Finally, in the introduction, we did not follow the referee comment in line 49.

*R: L.49: If the approach has been shown that it is a valid, saying that it has "not been further tested so far" is a bit odd. L.50: I would change "to the best of our knowledge, not been further tested so far" to something less absolute, like "has not been explored in depth" or something similar.*

We didn't change that part: we don't understand the referee request but we hope that the new structure of the introduction makes this part clearer to the referee. We underline that, from our point of view, this sentence is important for the message we want to communicate. However, we follow the referee advice for the subsequent line (L50) and we changed the text accordingly.

In the discussion paragraph, we deleted two sentences (lines 267-269 and lines 284-285 in the previous version of the manuscript): after reading the manuscript several times, we found these sentences redundant since the information was already given and clarified in the introduction.

We added the references included in Table 1 and lines 33-34 to the reference list.
We finally checked our manuscript carefully for typos, terminology, references.

All the variations in the text are in blue.

With kind regards
On behalf of all co-authors

Giulia Faucher